# Training Chain-of-Thought via Latent-Variable Inference

**Du Phan**[*]   **Matthew D. Hoffman**[*]   **David Dohan**[†]   **Sholto Douglas**   **Tuan Anh Le**
**Aaron Parisi**   **Pavel Sountsov**   **Charles Sutton**   **Sharad Vikram**   **Rif A. Saurous**
Google

## Abstract

Large language models (LLMs) solve problems more accurately and interpretably when instructed to work out the answer step by step using a "chain-of-thought" (CoT) prompt. One can also improve LLMs' performance on a specific task by supervised fine-tuning, i.e., by using gradient ascent on some tunable parameters to maximize the average log-likelihood of correct answers from a labeled training set. Naively combining CoT with supervised tuning requires supervision not just of the correct answers, but also of detailed rationales that lead to those answers; these rationales are expensive to produce by hand. Instead, we propose a fine-tuning strategy that tries to maximize the *marginal* log-likelihood of generating a correct answer using CoT prompting, approximately averaging over all possible rationales. The core challenge is sampling from the posterior over rationales conditioned on the correct answer; we address it using a simple Markov-chain Monte Carlo (MCMC) expectation-maximization (EM) algorithm inspired by the self-taught reasoner (STaR), memoized wake-sleep, Markovian score climbing, and persistent contrastive divergence. This algorithm also admits a novel control-variate technique that drives the variance of our gradient estimates to zero as the model improves. Applying our technique to GSM8K and the tasks in BIG-Bench Hard, we find that this MCMC-EM fine-tuning technique typically improves the model's accuracy on held-out examples more than STaR or prompt-tuning with or without CoT.

## 1   Introduction

For many mathematical, logical, and common-sense reasoning problems, large language models solve problems more accurately when instructed to work out the answer step by step in a *chain of thought* or a *scratchpad* (Wei et al., 2022; Nye et al., 2021; Kojima et al., 2022; Rajani et al., 2019; Shwartz et al., 2020). These methods encourage the model to produce a *rationale*, that is, text describing a sequence of reasoning steps that leads to an answer; the motivation is that it seems to be easier for the model to generate a sequence of correct reasoning steps than to generate the final answer directly. Because of the striking performance of chain-of-thought methods, many variants have been proposed (Wang et al., 2022b; Zhou et al., 2022; Creswell et al., 2022; Ye & Durrett, 2023), but there are still many cases in which the rationales are incorrect.

One way to improve these methods is to fine-tune models to generate better rationales. If gold-standard rationales can be obtained, such as via crowdsourcing (Rajani et al., 2019) or automatically (Nye et al., 2021), then supervised methods can be applied, but obtaining this data can be difficult. An appealing alternative is to start from datasets that contain questions and correct answers only, which are more readily available, and *bootstrap* rationales during learning. A version of this strategy

---

[*]Corresponding authors: `{mhoffman,phandu}@google.com`. The first two authors contributed equally, and their order was chosen randomly.

[†]Current affiliation: OpenAI.

37th Conference on Neural Information Processing Systems (NeurIPS 2023).

was proposed as the self-taught reasoner (STaR) (Zelikman et al., 2022), which generates proposed rationales from an LLM, and then fine-tunes on rationales that lead to the correct answer.

In this paper, we approach the problem of bootstrapping rationales from a different conceptual direction: *chain-of-thought methods are probabilistic latent-variable models*. The LLM defines a joint probability distribution over questions, rationales, and answers; this joint distribution implies a *marginal* distribution of answers given questions, averaging over all possible rationales weighted by their probability given the question. The problem of self-training for reasoning then becomes one of learning with incomplete data, a core task in probabilistic machine learning (Murphy, 2022) to which we can apply methods from a large and sophisticated literature.

This perspective raises a technical challenge, because computing the marginal distribution requires averaging over a vast set of potential rationales. To address this, we introduce a learning algorithm for rationale generation, which we call TRICE.[3] TRICE is a simple Markov-chain Monte Carlo (MCMC) expectation-maximization (EM) algorithm combined with a novel control-variate scheme, inspired by ideas from STaR (Zelikman et al., 2022), memoized wake-sleep (Hewitt et al., 2020), Markovian score climbing (Naesseth et al., 2020), and persistent contrastive divergence (Tieleman, 2008).

This view unifies several threads of work in reasoning using LLMs: It provides an alternative interpretation of STaR as a kind of biased stochastic expectation-maximization algorithm (Nielsen, 2000) that underweights difficult examples when its rationalization process fails. Self-consistency (Wang et al., 2022a) can be seen as a Monte Carlo algorithm for computing the most likely answer under the marginal distribution. Compared to self-consistency, the probabilistic learning approach of TRICE allows us to average over rationales not only at inference time, but also *at training time*. Compared to STaR, TRICE is less likely to ignore difficult examples (which stabilizes convergence and improves performance), and is also able to learn from *incorrect* rationales as well as correct ones.

We apply our technique to the GSM8K dataset (Cobbe et al., 2021) and to the BIG-Bench Hard benchmark (Suzgun et al., 2022a). We find that TRICE improves the model's performance significantly, outperforming models tuned with STaR, direct tuning with or without CoT, and even supervised fine-tuning on human-generated rationales.

## 2 Method

Given a training set of $N$ questions $x_{1:N}$ and answers $y_{1:N}$, we formalize CoT tuning as optimizing a parameter vector $\theta$ to maximize the average marginal log-likelihood of answers given questions:

$$\mathcal{L}(\theta) \triangleq \frac{1}{N} \sum_n \log p_\theta(y_n \mid x_n) = \frac{1}{N} \sum_n \log \sum_z p_\theta(z \mid x_n) p(y_n \mid z, x_n), \quad (1)$$

where $z$ is an unobserved latent rationale, $p_\theta(z \mid x)$ is the probability[4] of obtaining the rationale $z$ by prompting an LLM with the question $x$ and tunable parameters $\theta$, and $p_\theta(y \mid z, x)$ is the probability of obtaining the answer $y$ given rationale $z$, question $x$, and parameters $\theta$. We will be particularly interested in models where the likelihood $p_\theta(y \mid x, z) \in \{0, 1\}$, that is, where the answer $y$ is a deterministic function of $z$. For example, we might say that the model's answer is $y = $ "(a)" if $z$ ends with the string `"The answer is (a)."` For this deterministic model, we define $p(y \mid z, x) = c(z, y) \in \{0, 1\}$. Details of $c(z, y)$ for each task can be found in Appendix F. We believe that such a binary likelihood model is appropriate for question-answering tasks where $z$ is a rationale—a good rationale should leave no ambiguity about the correct answer. The derivations below will therefore assume a binary likelihood function. It is straightforward to generalize our methods to cases where the relationship between $z$ and $y$ is weaker and therefore $p(y \mid x, z)$ is more complicated; Appendix A shows how.

Algorithm 1 outlines the method. A notebook with a reference implementation can be found at `https://github.com/google-research/cascades/tree/main/cascades/examples/notebooks/trice.ipynb`.

We start by initializing a memory containing a latent rationale $z_n$ for each example pair $x_n, y_n$ by sampling $z_n$ from a hinted guide distribution $q(z \mid x_n, y_n)$ that may condition on the correct answer $y_n$ as well as the question $x_n$. For example, the guide might prompt an LLM specifically to give an

---

[3]TRICE stands for "Tuning Rationales with Independence-Chain Expectation-maximization."

[4]Unless otherwise specified, we sample at temperature 1 throughout.

rationale for the answer; more details about the precise prompts used by the guide are in Appendix F. In some cases sampling from the guide instead of the model $p_\theta(z \mid x_n)$ increases the chances of generating a correct rationale (Zelikman et al., 2022).

We then proceed to the main optimization loop. Each iteration, we sample a minibatch of $M$ questions and answers from the dataset, and retrieve the rationales associated with those examples from the memory. We then propose new rationales $\tilde{z}$ from the current model $p_\theta(z \mid x)$, and whenever the new rationale $\tilde{z}$ is correct (i.e., $c(\tilde{z}, y) = 1$) replace the old rationale in memory with the new one.

At this point we have all we need to compute a gradient estimate; we can just average the gradients $\nabla_\theta \log p_\theta(z_{i_m} \mid x_{i_m})$ that we obtain from those rationales in the updated memory that are correct (i.e., we ignore examples where both the proposed rationale and the previous rationale were wrong). BASIC_GRADIENT_ESTIMATE in Algorithm 1 shows how.

But we can also reduce the variance of our gradient estimator by incorporating a control variate, as in CONTROL_VARIATE_GRADIENT_ESTIMATE in Algorithm 1. We first compute leave-one-out estimates $\beta_{1:M}$ of the average probability of accepting a new rationale. For each example $m$, we subtract off a scaled control variate $\beta_m \nabla_\theta \log p_\theta(\tilde{z}_m \mid x_{i_m})$ whose expected value is zero (since it is a score function). If the proposed rationale $\tilde{z}_m$ for example $m$ is correct, then $z_{i_m} = \tilde{z}_m$, and the $m$th gradient contribution becomes $(1 - \beta_m)\nabla_\theta \log p_\theta(z_{i_m} \mid x_{i_m})$, i.e., it is scaled down by $1 - \beta_m$. If $\tilde{z}_m$ is incorrect, then we adjust the gradient estimate to try to make $\tilde{z}_m$ *less* likely under $p_\theta$. As the model becomes more accurate (i.e., $\beta$ gets closer to 1), we give more weight to incorrect rationales (when they occur) and less weight to correct rationales (most of the time).

CONTROL_VARIATE_GRADIENT_ESTIMATE is more expensive than BASIC_GRADIENT_ESTIMATE, since we must compute gradients not only for the rationales in memory but also for any incorrect rationales we generate. This may be wasteful, especially if many of the weights on those gradients ($1 - \beta$ for correct proposals, $\beta$ for incorrect proposals) are close to zero because $\beta$ is close to zero or one. To reduce this cost, in SUBSAMPLED_CONTROL_VARIATE_GRADIENT_ESTIMATE, we use systematic resampling (Hol et al., 2006) to generate a subsample of $L$ question-rationale pairs, from which we obtain an unbiased estimate of the output of CONTROL_VARIATE_GRADIENT_ESTIMATE. We preferentially sample gradients with higher scalar weights; if $\beta$ is small, we are less likely to sample incorrect rationales (which have weight $\beta$), and if $\beta$ is large, we are less likely to sample correct proposed rationales (which have weight $1 - \beta$). This can be seen as a generalization of the strategy of Burda et al. (2015, Section 3) for reducing the cost of computing IWAE gradients.

Below, we derive this variance-reduced stochastic MCMC-EM procedure in more detail.

## 2.1 Derivation

**The true gradient.** The gradient of the marginal log-likelihood $\log p_\theta(y \mid x)$ with respect to $\theta$ is

$$\nabla_\theta \log \sum_z p_\theta(z, y \mid x) = \sum_z \frac{p_\theta(z,y|x)\nabla_\theta \log p_\theta(z,y|x)}{\sum_{z'} p_\theta(z',y|x)} = \sum_z p_\theta(z \mid x, y)\nabla_\theta \log p_\theta(z \mid x), \quad (2)$$

that is, it is the expectation with respect to the posterior $p_\theta(z \mid x, y)$ of the gradient of the conditional log-prior $\log p_\theta(z \mid x)$, since the likelihood $p(y \mid z, x) = c(z, y)$ does not depend on $\theta$. So if we can sample from the posterior over rationales $z$ conditioned on the question-answer pair $x, y$, then we can compute an unbiased estimate of the gradient of the marginal log-likelihood $\log p_\theta(y \mid x)$. We can interpret this as "bootstrapping" rationales $z$ that are consistent with both the prior on rationales $p_\theta(z \mid x)$ and the observed answer $y$ (cf. Zelikman et al., 2022).

**Independence sampler for $p_\theta(z \mid x, y)$.** We cannot directly sample from $p_\theta(z \mid x, y)$, so we resort to Markov chain Monte Carlo (MCMC). We maintain a memory (cf. Hewitt et al., 2020) of a single rationale $z_n$ for each question-answer pair $x_n, y_n$, and each iteration we apply a random update to $z_n$ that leaves the posterior $p_\theta(z_n \mid x_n, y_n)$ invariant (cf. Tieleman, 2008). Each MCMC update brings the $z_n$'s closer in distribution to $p_\theta(z_n \mid x_n, y_n)$ (Cover, 1999; Murray & Salakhutdinov, 2008). However, updates to $\theta$ may change the posterior $p_\theta(z_n \mid x_n, y_n)$, so we must keep updating the chains to control the bias of our gradient estimates.

To update the chains, we use a simple, hyperparameter-free independence sampler (Tierney, 1994); a Metropolis-Hastings (Hastings, 1970) update that proposes updating the current state $z$ with a draw $\tilde{z}$ from a distribution $r_{x,y}$ that does not depend on $z$, and accepts the update with probability

---

**Algorithm 1** TRICE

---

**Input:** Generative model $p_\theta(z, y \mid x)$, is-correct function $c(z, y)$, dataset $x_{1:N}$, $y_{1:N}$, hinted guide distribution $q(z \mid x, y)$, initial parameters $\theta$, optimizer update function $h(\theta, g, t)$, minibatch size $M$, gradient minibatch size $L$, number of iterations $T$.
**Output:** Tuned parameters $\theta$, rationales $z_{1:N}$.

1: **for** $n \in 1, \ldots, N$ **do** (in parallel)      // Initialize Markov chain states.
2:    Sample $z_n$ from $q(z \mid x_n, y_n)$.     // Sample "fallback" rationale from guide $q$.
3: **end for**
4: **for** $t \in 1, \ldots, T$ **do**        // Main optimization loop.
5:    Get next minibatch of $M$ indices $i_{1:M}$ into the dataset.
6:    **for** $m \in 1, \ldots, M$ **do** (in parallel)    // Take one MCMC step to update Markov chain states.
7:      Sample $\tilde{z}_m$ from $p_\theta(z \mid x_{i_m})$.
8:      **if** $c(\tilde{z}_m, y_{i_m})$ **then**      // Accept or reject proposal.
9:        Update $z_{i_m} \leftarrow \tilde{z}_m$.
10:      **end if**
11:      Let $\tilde{c}_m = c(\tilde{z}_m, y_{i_m})$.      // Whether the proposal is correct.
12:      Let $c'_m = c(z_{i_m}, y_{i_m})$.      // Whether the updated rationale is correct.
13:    **end for**
14:    Compute $\hat{g}$ using either BASIC_GRADIENT_ESTIMATE$(z, x, c')$,
     CONTROL_VARIATE_GRADIENT_ESTIMATE$(z, x, \tilde{z}, \tilde{c}, c')$,
     or SUBSAMPLED_CONTROL_VARIATE_GRADIENT_ESTIMATE$(z, x, \tilde{z}, \tilde{c}, c')$.
15:    Update $\theta \leftarrow h(\theta, \hat{g}, t)$.      // Apply gradient update.
16: **end for**
17: **return** $\theta, z_{1:N}$.

18: **procedure** BASIC_GRADIENT_ESTIMATE$(z, x, c')$
19:    **return** $\frac{1}{\sum_m c'_m} \sum_m c'_m \nabla_\theta \log p_\theta(z_{i_m} \mid x_{i_m})$.
20: **end procedure**

21: **procedure** CONTROL_VARIATE_GRADIENT_ESTIMATE$(z, x, \tilde{z}, \tilde{c}, c')$
22:    **for** $m \in 1, \ldots, M$ **do** (in parallel)
23:      Set $\beta_m = \frac{\sum_{m' \neq m} c'_{m'} \tilde{c}_{m'}}{\sum_{m' \neq m} c'_{m'}}$.      // Compute leave-one-out control-variate scales.
24:    **end for**
25:    **return** $\frac{1}{\sum_m c'_m} \sum_m c'_m (\nabla_\theta \log p_\theta(z_{i_m} \mid x_{i_m}) - \beta_m \nabla_\theta \log p_\theta(\tilde{z}_m \mid x_{i_m}))$.
26: **end procedure**

27: **procedure** SUBSAMPLED_CONTROL_VARIATE_GRADIENT_ESTIMATE$(z, x, \tilde{z}, \tilde{c}, c')$
28:    **for** $m \in 1, \ldots, M$ **do** (in parallel)
29:      Set $\beta_m = \frac{\sum_{m' \neq m} c'_{m'} \tilde{c}_{m'}}{\sum_{m' \neq m} c'_{m'}}$.      // Compute leave-one-out control-variate scales.
30:      Set $\tilde{w}_m = c'_m(1 - \tilde{c}_m \beta_m)$,      // Compute unnormalized weights for subsampling.
       $\tilde{w}_{M+m} = c'_m(1 - \tilde{c}_m)\beta_m$.
31:    **end for**
32:    Choose a subset of $L$ indices $j_{1:L}$ using systematic resampling with probabilities $\frac{\tilde{w}}{\sum_{m=1}^{2M} \tilde{w}_m}$.
33:    **for** $\ell \in 1, \ldots, L$ **do** (in parallel)
34:      **if** $j_\ell \leq M$ **then**      // Selected correct rationale.
35:        Let $\hat{m} = j_\ell, \hat{z} = z_{i_{\hat{m}}}, s = 1$.
36:      **else**         // Selected incorrect rationale.
37:        Let $\hat{m} = j_\ell - M, \hat{z} = \tilde{z}_m, s = -1$.
38:      **end if**
39:      Compute $\hat{g}_\ell = s \nabla_\theta \log p_\theta(\hat{z} \mid x_{i_{\hat{m}}})$. // Negate gradient if $\ell$th rationale is incorrect.
40:    **end for**
41:    **return** $\frac{\sum_{m=1}^{2M} \tilde{w}_m}{\sum_m c'_m} \frac{1}{L} \sum_{\ell=1}^{L} \hat{g}_\ell$.
42: **end procedure**

---

$\alpha(\tilde{z} \mid z) = \min\left\{1, \frac{p_\theta(\tilde{z}, y \mid x)/r_{x,y}(\tilde{z})}{p_\theta(z, y \mid x)/r_{x,y}(z)}\right\}$. We choose $r_{x,y}(z) = p_\theta(z \mid x)$, which simplifies the acceptance probability to $\alpha(\tilde{z} \mid z) = \min\left\{1, \frac{p_\theta(y \mid x, \tilde{z})}{p_\theta(y \mid x, z)}\right\}$. This is 1 if $c(\tilde{z}, y) = 1$, 0 if $c(\tilde{z}, y) = 0$ and $c(z, y) = 1$, and ill-defined (implying that we have to reject) if both $c(z, y) = 0$ and $c(\tilde{z}, y) = 0$. So we accept whenever the proposal $\tilde{z}$ is correct, and reject otherwise.

*Remarks:* Independence samplers can be understood as "Metropolized" importance samplers that spread the work of generating and evaluating proposals over time. In our setting, the update can also be interpreted as attempting to sample from the posterior by rejection sampling, then falling back on an old sample if that fails. The expected number of iterations between successful updates is $p(y \mid x)^{-1}$, so mixing will be faster for easier questions $x$, and will accelerate as the model improves.

**Basic gradient estimator.** This MCMC/rejection-sampling procedure lets us approximate the gradient of the marginal log-likelihood in Equation (2). Denoting as $z$ the state[5] of the Markov chain for an example $x, y$ before the update, we sample a proposal $\tilde{z}$ from $p_\theta(z \mid x)$, accept the new state if it is correct (i.e., if $c(\tilde{z}, y) = 1$), and compute the gradient of the log-probability of the result:

$$z' = c(\tilde{z}, y)\tilde{z} + (1 - c(\tilde{z}, y))z; \quad \hat{g} = \nabla_\theta \log p_\theta(z' \mid x); \quad \mathbb{E}_{z,\tilde{z}}[\hat{g} \mid \theta] \approx \mathbb{E}_{p_\theta(z \mid x, y)}[\nabla_\theta \log p_\theta(z \mid x)], \tag{3}$$

where $\mathbb{E}_{z,\tilde{z}}[\cdot \mid \theta]$ denotes an expectation with respect to both the proposal $\tilde{z}$ and the previous state $z$.

*Remarks:* The estimate will have low bias if the distribution of $z'$ is close to the posterior $p(z \mid x, y)$, which we expect to be true if the chain is mixing quickly enough relative to how fast $\theta$ is changing. This will happen if either the probability of getting a correct answer is high, or if $\theta$ is changing slowly due to a small learning rate and/or gradient. If the model's training-set accuracy improves with training and we use a decaying learning-rate schedule, then as training proceeds both of these factors should work to reduce the bias of the gradient estimate.

**Adding a control variate.** The mean of an estimator $\hat{g}$ is not affected by subtracting a zero-mean random variable $b$ from it. If $b$ is positively correlated with $\hat{g}$, then $\hat{g} - b$ can have lower variance than $\hat{g}$, and we say that $b$ can be used as a "control variate" (Owen & Zhou, 2000). Since, by the score-function identity, $\mathbb{E}_{p_{z \mid x}}[\beta \nabla_\theta \log p_\theta(z \mid x)] = 0$ (for any scalar $\beta$ independent of $z$), we can use the proposed samples $\tilde{z}$ to generate control variates for our gradient estimator:

$$\begin{aligned} \mathbb{E}_{z,\tilde{z}}[\hat{g} \mid \theta] &= \mathbb{E}_z[\mathbb{E}_{\tilde{z}}[\nabla_\theta \log p_\theta(z' \mid x)] \mid \theta] \\ &= \mathbb{E}_z[\mathbb{E}_{\tilde{z}}[\nabla_\theta \log p_\theta(z' \mid x) - \beta \nabla_\theta \log p_\theta(\tilde{z} \mid x)] \mid \theta]. \end{aligned} \tag{4}$$

*Remarks:* The value of this estimator will depend on whether or not we accept the proposal $\tilde{z}$:

$$\begin{aligned} &\nabla_\theta \log p_\theta(z' \mid x) \\ &- \beta \nabla_\theta \log p_\theta(\tilde{z} \mid x) \end{aligned} = \begin{cases} (1 - \beta)\nabla_\theta \log p_\theta(z' \mid x) & \text{if } \tilde{c} = 1, \\ \nabla_\theta \log p_\theta(z' \mid x) - \beta \nabla_\theta \log p_\theta(\tilde{z} \mid x) & \text{if } \tilde{c} = 0, \end{cases} \tag{5}$$

where we use the shorthand $\tilde{c} \triangleq c(\tilde{z}, y)$.

This control variate can drive the variance of the gradient estimator to zero as the model converges to perfect accuracy on the training set (cf. Roeder et al., 2017). If we set $\beta = \pi$, where $\pi$ is the probability of a correct answer (i.e., that $\tilde{c} = 1$), then as $\pi$ gets large, most of the time $\tilde{c} = 1$ and we multiply our gradient estimator by $1 - \pi$ (multiplying its variance by a factor of $(1 - \pi)^2$). If $\tilde{c} = 0$, then we make use of both a correct and incorrect rationale; the weights attached to these updates will not be small, but if incorrect rationales are relatively rare then their contribution to the variance of the gradient estimator will be correspondingly small. On the other hand, if the model has not yet learned to frequently generate good rationales for the training examples, then we should set $\beta$ closer to 0, since in this case the signal from the incorrect rationale is less informative—in Appendix C.1 we show that the variance of gradient estimators based on incorrect rationales depends strongly on the model's accuracy $\pi$. In Appendix B, we show that choosing $\beta = \pi$ is in fact optimal up to $O((1 - \pi)^2)$ terms, and that the variance of the resulting estimator is proportional to $1 - \pi$.

**Estimating $\beta$.** For each example $x_m, y_m$, we need to compute a $\beta_m \approx \mathbb{E}[\tilde{c}_m]$ in a way that ensures that $\beta_m$ is independent of $\nabla_\theta \log p_\theta(\tilde{z}_m \mid x_m)$. We assume that $\mathbb{E}[\tilde{c}_m] \approx \frac{1}{M} \sum_m \mathbb{E}[\tilde{c}_m]$ (i.e., that the per-example acceptance probability is close to the average acceptance probability across the minibatch[6]), and compute the leave-one-out estimate $\beta_m = \frac{\sum_{m' \neq m} c'_{m'} \tilde{c}_{m'}}{\sum_{m' \neq m} c'_{m'}}$, where $c'_m := c(z'_m, y)$.

---

[5]There may be some examples (especially early in training) for which we have not yet generated any correct rationales. We omit these examples from our gradient estimate, since they have likelihood 0 and therefore cannot be representative samples from the posterior.

[6]We also tried keeping a running estimate of the average acceptance probability per example, but we did not find that this more complex scheme provided any empirical advantage.

We restrict the estimate to consider only examples for which we have a correct rationale (i.e., where $c'_m = 1$), since these are the only examples that influence our gradient estimate. Leaving out $\tilde{c}_m$ and $c'_m$ from the estimate $\beta_m$ ensures that $\beta_m$ is independent of $\tilde{z}_m$.

**Gradient subsampling.** Finally, as described above, we can reduce the cost of our gradient estimator by using systematic resampling to select a subset of rationales. This does not affect the expected value of the estimator as long as the marginal probability of selecting a rationale is proportional to the corresponding weight $\tilde{w}_m$, and the averaged gradient is reweighted by $\frac{\sum_{m=1}^{2M} \tilde{w}_m}{\sum_m c'_m}$.

## 2.2 Why not variational inference, reweighted wake-sleep, or rejection sampling?

We considered three alternatives to the MCMC-EM approach that we pursue in this paper: variational EM (e.g., Bishop, 2006), reweighted wake-sleep (RWS; Bornschein & Bengio, 2015; Le et al., 2019), and rejection sampling.

Variational expectation-maximization is a common strategy for training latent-variable models, but variational inference with discrete latent variables is challenging (e.g., Tucker et al., 2017).

RWS is an attractive alternative that avoids high-variance score-function gradients; it proceeds by sampling $M$ samples $z_{1:M}$ from a guide model $q_\phi(z \mid x, y)$, assigning the samples weights $w_m \propto \frac{p_\theta(y,z|x)}{q_\phi(z|x,y)}$, and updating both the model parameters $\theta$ and the guide parameters $\phi$ to maximize the reweighted log-probabilities $\sum_m w_m \log p_\theta(z_m \mid x)$ and $\sum_m w_m \log q_\phi(z_m \mid x, y)$. Unfortunately, we found that RWS training sometimes led to degenerate zero-length rationales $z$. Figure 1 suggests a partial explanation: shorter sequences get higher weights, so the model and guide learn to produce shorter and shorter sequences until they consistently produce empty rationales.

Why do longer sequences tend to get lower weights? We can write the unnormalized weights as $\tilde{w}_m = c(y, z_m) \frac{p_\theta(z_m|x)}{q_\phi(z_m|x,y)} = c(y, z_m) \prod_{t=1}^{T_m} \frac{p_\theta(z_{m,t}|x,z_{m,1:(t-1)})}{q_\phi(z_{m,t}|x,y,z_{m,1:(t-1)})}$, where $T_m$ is the length of $z_m$ and $\epsilon$ is added to address the case where none of the samples are correct. If there is a mismatch between $q(z_{m,t} \mid x, z_{m,1:(t-1)}))$ and $p(z_{m,t} \mid x, z_{m,1:(t-1)})$, then $\frac{p_\theta(z_{m,t}|x,z_{m,1:(t-1)})}{q_\phi(z_{m,t}|x,y,z_{m,1:(t-1)})}$ will usually be less than one, with rare high-weight exceptions that ensure that $\mathbb{E}_q[p(z \mid x)/q(z \mid x)] = 1$.

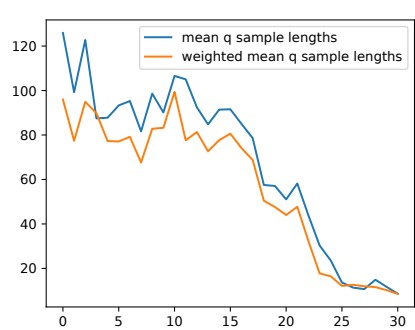

Figure 1: Example of rationale lengths shrinking during RWS training. Blue line shows the average number of tokens per rationale generated by the guide, orange line shows the average number of tokens per rationale weighted by the rationale's importance weight.

If these exceptions are rare enough to not typically appear in a sample of $M$ sequences $z_{1:M}$, then the normalized weights $w_{1:M} = \frac{\tilde{w}_{1:M}}{\sum_m \tilde{w}_m}$ will tend to assign higher mass to shorter sequences unless those shorter sequences are much less likely to be correct.

With careful initialization and learning-rate tuning, we could sometimes get RWS to avoid this problem of empty rationales. But this led to a new problem: the guide $q_\phi(z \mid x, y)$ learned to closely mimic the prior $p(z \mid x)$ until the very end of the rationale, and then simply paste in the correct answer whether or not it had anything to do with the rationale up to that point (cf. Turpin et al., 2023). Figure 5 in Appendix E shows a representative example in which the guide model ignores the answer it arrived at through incorrect reasoning and pastes in the correct answer.

Quantitatively, denoting by $t$ the index of the token at which the "final answer" section of the rationale begins, in one run we found that the average KL between $q(z_{1:t} \mid x, y)$ and $p(z_{1:t} \mid x)$ was about $0.61$ nats, while the conditional KL between $q(z_{(t+1):T} \mid x, y, z_{1:t})$ and $p(z_{(t+1):T} \mid x, z_{1:t})$ was about $42.5$ nats, confirming that the guide was not "reasoning backwards", just copying the correct answer.

Finally, we considered a rejection-sampling[7] scheme in which we sample $K$ proposal rationales $z_{1:K}$ from $p(z \mid x)$, and average the gradients from those rationales that lead to correct answers. We will present the quantitative results in Section 4; our main finding is that, while this scheme can work, it requires reducing the minibatch size by a factor of $K$ to keep the per-iteration cost constant compared to TRICE, which in turn leads to slower convergence and/or worse final results.

# 3 Related Work

A number of methods have proposed rationale generation for problem-solving tasks in neural sequence models, including both fully supervised and few-shot approaches (Wei et al., 2022; Nye et al., 2021; Kojima et al., 2022; Rajani et al., 2019; Shwartz et al., 2020; Wang et al., 2022b; Zhou et al., 2022; Creswell et al., 2022; Ye & Durrett, 2023). Particularly relevant to our approach is self-consistent chain-of-thought (Wang et al., 2022b), because this can be approximately viewed as marginalizing over rationales at test time. This technique has been successfully applied for a range of quantitative reasoning tasks (Lewkowycz et al., 2022). There is relatively much less work that does imputation or averaging over rationales at training time; perhaps the main instance is STaR (Zelikman et al., 2022), which we discuss in Section 3.1.

Dohan et al. (2022) present a position paper which advocates representing a composition of language model interactions via probabilistic programming. Our treatment of rationales as latent variables is inspired by that work. Lievin (2022) offers another example of interpreting LLMs with CoT as latent-variable models.

Variational inference (e.g., Kingma & Welling, 2013) and wake-sleep methods (e.g., Bornschein & Bengio, 2015) are workhorses of the latent-variable-modeling community, but as we discuss in Section 2.2 we found the bias of these methods to cause serious problems. MCMC-EM is a less-common strategy these days, although a version of it based on Gibbs sampling (Geman & Geman, 1984) it has been widely applied to training undirected graphical models (Tieleman, 2008). TRICE can also be cast as an instance of Markovian score climbing (Naesseth et al., 2020).

ReAct (Yao et al., 2023) demonstrated that injecting reasoning into an RL-style observe-and-act loop significantly increases performance. This approach was extended in Reflexion (Shinn et al., 2023), where an agent can conditionally reflect on an RL trajectory, augmenting the resulting examples which can be used as few-shot examples in subsequent rollouts. These approaches reported significant improvements on their respective evaluation tasks but still rely on the model being able to produce useful and actionable feedback through pure few-shot prompting, whereas our method actively tunes the model to produce thoughts amenable to the task.

Recent work on tool-use within language models also works via imputation, inferring where to insert calls to tools (Parisi et al., 2022; Schick et al., 2023). Their loss functions are similar in spirit to ours, filtering out trajectories which do not lead to valid answers. In this paper, we have treated rationales as latent variables; one could also treat tool-use as a latent variable.

## 3.1 Self-Taught Reasoner

The most closely related work is the self-taught reasoner (STaR; Zelikman et al., 2022). Besides the arguments in their derivations, there are three significant differences between TRICE and STaR. First, STaR uses greedy decoding, which reduces the diversity of the rationales it trains on. The authors made this choice to reduce the danger of the model getting the right answer despite having a bad rationale. While we do find that our procedure sometimes generates correct answers for the wrong reasons, this did not seem to stand in the way of the model improving on most tasks. One reason may be that our base models are more powerful than the 6B-parameter GPT-J model used in the STaR paper, so they are more likely to generate good rationales from the beginning.

A second difference is that TRICE resamples rationales every iteration, so it are less likely to overfit to any particular rationale. STaR has an inner loop that runs many training iterations on a single set of rationales, meaning it uses stale rationales to estimate the gradient of the marginal likelihood.

---

[7]We also considered optimizing an importance-weighted bound (Burda et al., 2015) using the prior $p(z \mid x)$ as a proposal distribution, but instead opted for a simple rejection sampling scheme since this is less biased and equally feasible in our setting.

In our experiments, we observed that this leads to the model effectively memorizing a fixed set of rationales for the training set. Once this happens, the greedy decoding procedure will almost certainly reproduce exactly the same rationales at the beginning of the next outer loop. If these rationales all lead to the correct answer, and STaR has a rationale for each question, then this is a global optimum of the marginal likelihood on the training set! But empirically, STaR often does not find a good rationale for each question, and so it ignores some fraction of the training set (see Section 4).

This tendency to ignore the most difficult questions in the training set follows from STaR's derivation as an approximate policy-gradient algorithm trying to directly minimize the 0-1 loss $\mathbb{E}_p[1 - c(z, y)] = 1 - p_\theta(y \mid x)$. The derivative of this marginal likelihood is $p_\theta(y \mid x)\nabla_\theta \log p_\theta(y \mid x)$, that is, it is the derivative of the marginal *log*-likelihood (which TRICE tries to maximize) *weighted by* $p_\theta(y \mid x)$. This weighting causes difficult examples to contribute little to the gradient used to update the model, so the model may "give up" on questions that it cannot yet solve. This is one argument for trying to maximize log-likelihoods instead of likelihoods.

A final, minor difference is that when STaR updates its rationales, it may replace a rationale from the model $p(z \mid x)$ with a rationale from a surrogate $q_\theta(z \mid x, y)$. As the model memorizes a set of correct rationales for the training set, STaR becomes less likely to fall back on the surrogate, but this choice could affect early training dynamics.

# 4   Experiments

We evaluate TRICE on the GSM8K (Cobbe et al., 2021) dataset and the 27 BigBench-Hard (BBH) tasks (Suzgun et al., 2022b) using the medium-size PaLM 2-M (Anil et al., 2023) Transformer-based LLM (Vaswani et al., 2017). For the BBH experiments, we used the Flan instruction-tuned (Chung et al., 2022) version of PaLM 2; for GSM8K, we used the base PaLM 2 model, since GSM8K is included in the Flan training datasets. All experiments were run on TPU v4 and v5e chips (Jouppi et al., 2023). Examples of generated rationales can be found in Appendix E.

Rather than fine-tune the model weights, we use *prompt tuning* (Lester et al., 2021); we prepend a sequence of embedding vectors $\theta$ (a "soft prompt") to the embeddings corresponding to the tokenized CoT prompt used to condition the model. Prompt tuning can achieve similar accuracy gains to full fine-tuning, but using a small fraction of the parameters. We initialize the soft prompt with the embedding sequence obtained from a series of three (for BBH) or five (for GSM8K) exemplar CoT prompts, each of the form "`Question:  <QUESTION>\nAnswer:  Let's think step by step.\n<RATIONALE>`". We consider two initialization schemes: one where we use the standard few-shot CoT prompts that are provided with BBH, and one where we try to bootstrap a few-shot CoT prompt by sampling random questions from the training set, generating random rationales from the base model, and picking three or five examples where the random rationales lead to correct answers. The first scheme can be seen as a way of fine-tuning a good initial few-shot prompt, but it does require a small amount of detailed CoT supervision, while the second scheme only requires label supervision.

On each BBH task, we split the examples into 60% train and 40% test sets. For all but three tasks, this is 150 training and 100 test examples. For GSM8K, we use the standard 7473-example training set and 1319-example test set. We evaluate CoT models' accuracy in two ways: first, using greedy (temperature-0) decoding, and second, using "self-consistency" (Wang et al., 2022b). In self-consistency evaluation, we draw 40 samples and check whether the most common answer is correct; this is a plug-in estimator for the prediction $\arg\max_y p(y \mid x)$ that minimizes 0-1 loss under the model (although this is not how Wang et al. (2022b) originally motivated the procedure).

We compare against four baseline prompt-tuning methods: direct prompt tuning, CoT prompt tuning, rejection sampling, and STaR (Zelikman et al., 2022). All methods are evaluated against the same validation sets, and use the same training labels, few-shot prompts (except for direct tuning, where we only use question-answer pairs), and initialization strategies as appropriate. Details for each method and its corresponding experimental hyperparameters can be found in Appendix F.

Table 1 and Table 2 summarize the results; more detailed task-by-task BBH summaries are in Appendix D. Even with no human-generated exemplar rationales, TRICE is able to learn to generate rationales that lead to the correct answer. TRICE also outperforms a model trained directly on human-generated rationales on GSM8K (cf. Uesato et al., 2022), perhaps because the cross-entropy

| Prompt-Tuning Strategy | Initialization | Greedy-Decoding Acc. (%) | Self-Consistency Acc. (%) | % Valid Rationales |
|---|---|---|---|---|
| STaR | Bootstrapped 3-shot Q-R-A | 62.0 | 62.1 | 91.6 |
| Rejection Sampling | | 64.6 | 65.3 | - |
| TRICE without CV | | 67.8 | 68.0 | 98.7 |
| TRICE with CV | | **72.8** | **73.1** | **98.8** |
| Direct Prompt Tuning | 3-shot Q-A | 70.4 | - | - |
| TRICE without CV | 3-shot Q-R-A | 73.4 | 75.2 | 98.2 |
| TRICE with CV | | **76.7** | **77.6** | **98.6** |

Table 1: Average accuracies (columns 3 and 4) and fraction of training examples for which we can generate correct rationales (column 5) across the 27 BIG-Bench Hard (BBH) tasks. All methods but direct prompt tuning use CoT prompting. All trainable prompts are initialized with an embedding sequence obtained from a few-shot prompt containing either example question-answer pairs ("Q-A") or example question-rationale-answer triples ("Q-R-A"). For direct prompt tuning, the Q-A pairs come from the training set. For TRICE, we use either the three Q-R-A triples provided with BBH (bottom two rows) or bootstrap a set of rationales as described in the text. For STaR and rejection sampling, we only evaluate on bootstrapped initializations.

| Prompt-Tuning Strategy | Greedy-Decoding Acc. (%) | Self-Consistency Acc. (%) | % Valid Rationales |
|---|---|---|---|
| STaR | 53.5 | 60.1 | 80.2 |
| CoT Prompt Tuning | 58.6 | 73.8 | - |
| Rejection Sampling | **77.9** | **87.0** | - |
| Direct Prompt Tuning | 19.4 | - | - |
| TRICE without CV | 72.8 | 81.5 | **98.9** |
| TRICE with CV | 74.7 | 82.3 | 98.8 |
| TRICE with CV (not bootstrapped) | 77.7 | 86.6 | 98.4 |

Table 2: Average accuracies (columns 2 and 3) and fraction of training examples for which we can generate correct rationales (column 4) on GSM8K. Direct prompt tuning is initialized with an embedding sequence obtained from a few-shot prompt containing example question-answer pairs ("Q-A"). All remaining prompt-tuning methods are initialized with an embedding sequence obtained from a few-shot prompt containing example question-rationale-answer triples ("Q-R-A") obtained randomly from the GSM8K training set or bootstrapped as described in the text.

loss used in supervised fine-tuning may place more weight on style than substance; it takes far more bits to encode how one *expresses* a chain of reasoning than it does to encode the reasons themselves.

Initializing the soft prompt with a human-generated 3-shot exemplar question-rationale-answer prompt slightly improves performance on BBH, as does evaluating with self-consistency. By the end of training, TRICE has managed to generate at least one valid rationale for almost all training examples, while STaR fails to generate valid rationales for almost 10% of training examples. Unlike in the experiments done on Commonsense QA (Talmor et al., 2019) by Zelikman et al. (2022), STaR does not outperform the direct-prompted prompt-tuned model on BBH. This may be because each BBH task includes relatively little training data (150 examples as opposed to CommonsenseQA's 9,741), and so in its inner loop STaR overfits to its relatively small set of bootstrapped rationales. TRICE, on the other hand, can overfit to the small set of *questions* but at least has a chance to generate a somewhat diverse set of *rationales* from those questions.

One piece of evidence for this overfitting-rationales hypothesis is that on the final step of its final inner loop, STaR (with bootstrapped initialization) achieves a training sequence-level (*not* per-token) cross-entropy loss of less than 0.06 on all tasks, and of less than 0.01 on 19 out of 27 tasks. This implies that it has learned to exactly reproduce a single set of rationales with very high probability, which makes it very likely that it will generate those same rationales in the next iteration.

Figure 2 compares estimates for GSM8K of the average training marginal likelihood (i.e., how often a proposal is accepted) and the validation accuracy with greedy decoding as a function of number of

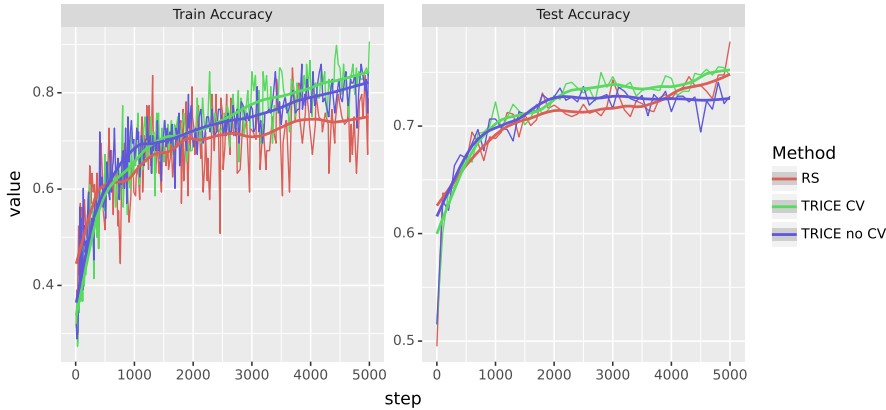

Figure 2: Time-varying estimates (with loess smoothers) of average training-set accuracy $p(y \mid x)$ and greedy-decoding validation-set accuracy for TRICE with and without subsampled control-variate gradient estimator ("TRICE CV" and "TRICE no CV" respectively) and four-particle rejection sampling ("RS") on GSM8K.

training steps[8] for rejection sampling and for TRICE with and without the control-variate scheme. The control-variate scheme improves average convergence speed, particularly towards the end of training as the probability of generating correct answers on the training set increases. Both versions of TRICE converge to high training accuracy much faster than rejection sampling.

## 5   Discussion

We proposed TRICE, a method for tuning LLMs to be better at solving question-answering tasks using chain-of-thought (CoT) prompting. By framing the CoT-prompted LLM as a latent-variable model, we were able to derive a principled and effective fine-tuning method. When applied to GSM8K and BIG-Bench Hard (BBH) tasks, TRICE outperforms three strong baselines: direct prompt-tuning, STaR, and rejection sampling. While we derived TRICE in the context of CoT question-answering, its basic MCMC-EM strategy could be employed more broadly, for example to tool-use problems.

**Limitations:**   We only evaluated TRICE with prompt-tuning on a medium-size LLM; it may be that it behaves differently on smaller models, larger models, or when using other fine-tuning strategies. TRICE is a gradient-based tuning algorithm, but many of the most capable LLMs are proprietary, and their owners often do not provide any public mechanism for gradient-based fine-tuning. This makes it hard to evaluate how well TRICE would work when used with, say, GPT-4 (OpenAI, 2023). Finally, our quantitative evaluations focused on whether the LLM could produce the right answer; we did not formally evaluate the quality of the reasoning in the rationales themselves (cf. Uesato et al., 2022).

**Broader Impacts:**   This work aims to improve the capabilities of LLMs by making them better able to answer questions accurately and transparently. However, more-capable LLMs may be used in malicious or unsafe ways, fine-tuning on uncurated question-answering datasets may introduce biases into the models, and more widely used LLMs will contribute a larger carbon footprint.

Rationales may make it easier for motivated users to judge the trustworthiness of LLM outputs. But many users may not read and critique an LLM's rationales, taking the mere existence of a rationale as evidence of truth. If chain-of-thought rationales promote uncritical trust, they could lead to harm.

---

[8]We set the cost per iteration of rejection sampling and TRICE with and without the control-variate scheme to be directly comparable: for rejection sampling, we reduce the minibatch size by a factor of four and generate four times as many proposals per example; for TRICE with the control-variate scheme, we set the gradient minibatch size $L$ equal to the number of examples per minibatch $M$ (note that this does still involve subsampling, since each example could potentially contribute both a correct and an incorrect rationale to the gradient estimate).

**Acknowledgements:** We appreciate Daniel Freeman and Enrique Piqueras' contributions to the infrastructure that we used in our experiments. We thank Kevin Murphy, Ben Lee, Brian Patton, and Jascha Sohl-Dickstein for helpful discussions.

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

# Supplemental Material for "Training Chain-of-Thought via Latent-Variable Inference"

## A  Generalizing TRICE to Nondeterministic Likelihood Models

To apply TRICE beyond question-answering problems, we might want to use a nondeterministic likelihood model. For example, our desired output $y$ might be a summary of a text $x$, and $z$ might be a scratchpad or outline. In situations like this, there might be many $y$'s that are appropriate for a given $x$ and $z$. So from a modeling perspective, it could make sense to make $p(y \mid x, z)$ have nonzero entropy. But there is also a computational reason to prefer such a model: as the number of reasonable values that $y$ could take given $x$ increases, the probability of sampling the precise $y$ that was observed goes down at a rate that might be exponential in the size of the output space.

Fortunately, we can easily extend TRICE to accommodate nondeterministic likelihoods. The differences are:

1. As discussed in Section 2.1, the acceptance probability for an independence sampler where the proposals are generated from $p(z \mid x)$ is $\alpha(\tilde{z} \mid z) = \min\{1, \frac{p(y \mid \tilde{z}, x)}{p(y \mid z, x)}\}$. So instead of deterministically accepting correct proposed latents $\tilde{z}$, we update the memory probabilistically, always accepting proposals that make the observed $y$ more likely, but sometimes also accepting proposals that make $y$ a little less likely.

2. The control-variate weights $\beta_m$ are now a function of the acceptance probabilities $\alpha$: $\beta_m = \frac{1}{M-1} \sum_{m' \neq m} \alpha(\tilde{z}_{m'} \mid z_{i_{m'}})$.

The value of the control variate in this setting may be less than it is in the deterministic-likelihood setting. Even if we learn a model that consistently produces good latents $z$ (in the sense that they lead to valid outputs $y$), this does not guarantee that it will consistently generate latents that are consistent with the *particular* $y$ that was observed. For example, there might be multiple reasonable ways to outline a long text, some of which lead to different summary paragraphs. In this scenario, the acceptance probability may not converge to something close to 1, and the variance-reduction effect from the control variate will be modest.

## B  Derivation of the Control Variate Scaling Heuristic

Our analysis will focus on the variance of the $k$th element $\hat{g}_k$ of our gradient estimator. To minimize clutter, we suppress subscript indices and define $g' = \nabla_{\theta_k} \log p(z' \mid x)$, $\tilde{g} = \nabla_{\theta_k} \log p(\tilde{z} \mid x)$, and $g = \nabla_{\theta_k} \log p(z \mid x)$, where $z'$ is the updated rationale, $\tilde{z}$ is the proposed rationale, and $z$ is the previous rationale in memory. Note that $g' = \tilde{g}$ if the proposal $\tilde{z}$ is accepted, and $g' = g$ if $\tilde{z}$ is rejected. We assume that the previous rationale $z$ is a sample from $p(z \mid x, y)$. We also define the shorthands $g_+ = \mathbb{E}[\tilde{g} \mid c(\tilde{z}, y) = 1]$, $g_- = \mathbb{E}[\tilde{g} \mid c(\tilde{z}, y) = 0]$, $v = \mathrm{Var}(\tilde{g}) = \mathbb{E}[\tilde{g}^2]$, $v_+ = \mathrm{Var}(\tilde{g} \mid c(\tilde{z}, y) = 1)$, $v_- = \mathrm{Var}(\tilde{g} \mid c(\tilde{z}, y) = 0)$. Finally, we define the probability of accepting a proposal $\pi = \mathbb{E}[c(\tilde{z}, y)]$.

Our gradient estimator is

$$\hat{g} = g' - \beta\tilde{g}; \quad \mathbb{E}[\hat{g}] = g_+. \tag{6}$$

The variance of $\hat{g}$ is

$$\mathbb{E}[(g' - g_+ - \beta\tilde{g})^2] = \mathbb{E}[(g' - g_+)^2] + \beta^2\mathbb{E}[\tilde{g}^2] - 2\beta\mathbb{E}[(g' - g_+)\tilde{g}] \tag{7}$$

$$= v_+ + \beta^2 v - 2\beta\mathbb{E}[g'\tilde{g}]. \tag{8}$$

In Equation (8) we use the fact that $\mathbb{E}[\tilde{g}] = 0$ and $g_+$ is nonrandom and therefore independent of $\tilde{g}$. Breaking out the cases where $c(\tilde{z}, y) = 0$ (so $g' = g$) and $c(\tilde{z}, y) = 1$ (so $g' = \tilde{g}$), the rightmost expectation is

$$\mathbb{E}[g'\tilde{g}] = \pi\mathbb{E}[\tilde{g}^2 \mid c(\tilde{z}, y) = 1] + (1 - \pi)\mathbb{E}[g\tilde{g} \mid c(\tilde{z}, y) = 0] \tag{9}$$

$$= \pi(v_+ + g_+^2) + (1 - \pi)g_+ g_-. \tag{10}$$

We can simplify the term on the right. Since $0 = \mathbb{E}[\tilde{g}] = \pi g_+ + (1-\pi)g_-$, $(1-\pi)g_+g_- = -\pi g_+^2$. Plugging this into Equation (10), we have

$$\mathbb{E}[g'\tilde{g}] = \pi(v_+ + g_+^2) - \pi g_+^2 = \pi v_+. \tag{11}$$

So the variance of $\hat{g}$ simplifies to

$$\mathrm{Var}(\hat{g}) = (1 - 2\beta\pi)v_+ + \beta^2 v. \tag{12}$$

Taking the derivative with respect to $\beta$ shows that this is minimized when

$$2(\beta v - \pi v_+) = 0; \quad \beta^\star = \pi\frac{v_+}{v}. \tag{13}$$

Plugging this back into Equation (12) gives the optimal variance $v^\star$:

$$v^\star = v_+ + \pi^2\frac{v_+^2}{v} - 2\pi^2\frac{v_+^2}{v} = v_+\left(1 - \pi^2\frac{v_+}{v}\right). \tag{14}$$

We can expand $v$ by considering the case where $c(\tilde{z}, y) = 1$ (which happens with probability $\pi$ and the case where $c(\tilde{z}, y) = 0$ (which happens with probability $1 - \pi$):

$$
\begin{aligned}
v = \mathbb{E}[\tilde{g}^2] &= \pi v_+ + \pi g_+^2 + (1-\pi)v_- + (1-\pi)g_-^2 \\
&= \pi v_+ - (1-\pi)g_+g_- + (1-\pi)v_- - \pi g_+g_- \\
&= \pi v_+ + (1-\pi)v_- - g_+g_- \\
&= \pi v_+ + (1-\pi)v_- + \frac{1-\pi}{\pi}g_-^2 \\
&= \pi v_+ + (1-\pi)v_- + (1-\pi)(2-\pi)g_-^2 + O((1-\pi)^3) \\
&= \pi v_+ + (1-\pi)v_- + (1-\pi)(1+1-\pi)g_-^2 + O((1-\pi)^3) \\
&= \pi v_+ + (1-\pi)(v_- + g_-^2) + O((1-\pi)^2),
\end{aligned}
\tag{15}
$$

where in the second line we again use the fact that $\pi g_+ = -(1-\pi)g_-$, and in the third-to-last line we approximate $\frac{1}{\pi}$ with the first-order Taylor approximation $\frac{1}{\pi} = 2 - \pi + O((1-\pi)^2)$. Thus, we can write

$$
\begin{aligned}
\pi\frac{v_+}{v} &= \frac{\pi v_+}{\pi v_+ + (1-\pi)(v_- + g_-^2) + O((1-\pi)^2)} \\
&= \left(1 + \frac{1-\pi}{\pi v_+}(v_- + g_-^2) + O((1-\pi)^2)\right)^{-1} \\
&= 1 - \frac{1-\pi}{\pi v_+}(v_- + g_-^2) + O((1-\pi)^2).
\end{aligned}
\tag{16}
$$

So the optimal variance $v^\star$ is

$$
\begin{aligned}
v^\star &= v_+\left(1 - \pi^2\frac{v_+}{v}\right) \\
&= v_+\left(1 - \pi + \frac{1-\pi}{v_+}(v_- + g_-^2) + O((1-\pi)^2)\right) \\
&= (1-\pi)(v_+ + v_- + g_-^2) + O((1-\pi)^2)).
\end{aligned}
\tag{17}
$$

By contrast, plugging our heuristic value of $\beta = \pi$ into Equation (12) gives the suboptimal variance $v^\pi$:

$$
\begin{aligned}
v^\pi &= (1 - 2\pi^2)v_+ + \pi^2 v \\
&= (1 - 2\pi^2)v_+ + \pi^2(\pi v_+ + (1-\pi)(v_- + g_-^2) + O((1-\pi)^2)) \\
&= (1 - 2\pi^2 + \pi^3)v_+ + \pi^2(1-\pi)(v_- + g_-^2) + O((1-\pi)^2) \\
&= (1-\pi)(v_+ + v_- + g_-^2) + O((1-\pi)^2),
\end{aligned}
\tag{18}
$$

where we use the approximation $\pi^k = (1 - (1-\pi))^k = 1 - k(1-\pi) + O((1-\pi)^2)$ to simplify the $\pi^2$ and $\pi^3$ terms. Thus, we conclude that $v^\star$ and $v^\pi$ are the same up to $O((1-\pi)^2)$, and so as the probability $\pi$ of getting the correct answer increases, the suboptimality of setting $\beta = \pi$ goes down faster than the variance does.

# C   On Gradient Estimators Based Solely on Incorrect Rationales

We adopt the same shorthands as in Appendix B.

Our analysis centers on the identity

$$0 = \mathbb{E}[\tilde{g}] = \pi g_+ + (1 - \pi)g_-, \tag{19}$$

which relates the gradient we want to estimate $g_+$ (the expected gradient given that the rationale is correct) to $g_-$ (the expected gradient given that the rationale is incorrect).

## C.1   Variance of incorrect-rationale gradient estimators

Equation (19) implies that

$$g_+ = -\frac{1 - \pi}{\pi}g_-. \tag{20}$$

Deferring for the moment the difficulty in estimating $\pi^{-1}$ (see Appendix C.2 below), we can consider the variance of an estimator based on the right hand side of Equation (20):

$$
\begin{aligned}
\mathbb{V}\left[-\frac{1 - c(\tilde{z}, y)}{\pi}\tilde{g}\right] &= \mathbb{E}\left[\frac{(1 - c(\tilde{z}, y))^2}{\pi^2}\tilde{g}^2\right] - g_+^2 \\
&= \frac{1 - \pi}{\pi^2}(v_- + g_-^2) - g_+^2 \\
&= \frac{1 - \pi}{\pi^2}(v_- + g_-^2) - \frac{(1 - \pi)^2}{\pi^2}g_-^2 \\
&= \frac{1 - \pi}{\pi^2}(v_- + \pi g_-^2).
\end{aligned}
\tag{21}
$$

If $\pi$ is small, then this becomes

$$\frac{1 - \pi}{\pi^2}(v_- + \pi g_-^2) = \frac{v_-}{\pi^2} + O(\pi^{-1}), \tag{22}$$

so that unless the variance $v_-$ of incorrect rationales is very low, the variance of this estimator will be $O(\pi^{-2})$, which is very high. By contrast, the variance of a gradient estimator based purely on correct rationales is simply $v_+$, so unless the gradient variance for incorrect rationales is dramatically lower than that for correct rationales, then if $\pi$ is small then incorrect rationales will lead to much noisier gradient estimates.

On the other hand, if $1 - \pi$ is small, then we have

$$\frac{1 - \pi}{\pi^2}(v_- + \pi g_-^2) = (1 - \pi)(v_- + g_-^2) + O((1 - \pi)^2), \tag{23}$$

which is likely a significant improvement on the variance $v_+$ of the correct-rationale estimator; in particular, it goes to zero as $\pi$ approaches 1.

## C.2   TRICE control variate as a debiased estimator based on incorrect rationales

Equation (20) implies that, in principle, given unbiased estimates for $1 - \pi$ and $\pi^{-1}$, we could compute gradient updates purely based on rationales that *fail* to obtain the correct answer. Unbiased estimates of $1 - \pi$ are easy to obtain; we need only compute $1 - c(\tilde{z}, y)$. But unbiased estimates of $\pi^{-1}$ are harder to come by.

Instead, we can compute a *biased* estimator that ignores the $\pi^{-1}$ term and then correct for the bias:

$$
\begin{aligned}
\mathbb{E}[-(1 - c(\tilde{z}, y))\tilde{g}] &= -(1 - \pi)g_- \\
&= \pi g_+ \\
&= g_+ - (1 - \pi)g_+.
\end{aligned}
\tag{24}
$$

So the expected value of the estimator $-(1 - c(\tilde{z}, y))\tilde{g}$ is too small by a factor of $(1 - \pi)g_+$. We can correct this bias by adding in an unbiased estimator of it that uses $g$ (the gradient for a correct rationale from memory):

$$\mathbb{E}[-(1 - c(\tilde{z}, y))(\tilde{g} - g)] = g_+ - (1 - \pi)g_+ + (1 - \pi)g_+ = g_+. \tag{25}$$

This is precisely the gradient estimator that TRICE uses when the sampled rationale $\tilde{g}$ is incorrect and $\beta = 1$. Smaller values of $\beta$ interpolate between this estimator and an estimator based purely on correct rationales.

| Task | Bootstrapped 3-shot Q-R-A | | | | 3-shot Q-R-A | | 3-shot Q-A |
|---|---|---|---|---|---|---|---|
| | TRICE (no CV) GD / SC | TRICE (CV) GD / SC | STaR GD / SC | RS-4 GD / SC | TRICE (no CV) GD / SC | TRICE (CV) GD / SC | Direct Tune GD |
| boolean_expressions | 94.0 / 94.0 | 92.0 / 93.0 | 88.0 / 89.0 | 91.0 / 92.0 | 89.0 / 92.0 | **96.0 / 95.0** | 85.0 |
| causal_judgement | 57.3 / **64.0** | 61.3 / 61.3 | **66.7** / 62.7 | 52.0 / 53.3 | 57.3 / 58.7 | 61.3 / 57.3 | 46.7 |
| date_understanding | 73.0 / 74.0 | 71.0 / 73.0 | 64.0 / 63.0 | 64.0 / 64.0 | 86.0 / 85.0 | **87.0 / 86.0** | 79.0 |
| disambiguation_qa | 96.0 / 96.0 | **97.0 / 97.0** | 83.0 / 82.0 | **97.0 / 97.0** | 89.0 / 90.0 | **97.0 / 97.0** | 81.0 |
| dyck_languages | **90.0** / 89.0 | 85.0 / 87.0 | 64.0 / 65.0 | 89.0 / **90.0** | 51.0 / 60.0 | 68.0 / 70.0 | 50.0 |
| formal_fallacies | 49.0 / 53.0 | **66.0 / 65.0** | 55.0 / 53.0 | 47.0 / 45.0 | 52.0 / 59.0 | 41.0 / 46.0 | 52.0 |
| geometric_shapes | **99.0 / 99.0** | 92.0 / 94.0 | 85.0 / 85.0 | 48.0 / 48.0 | 92.0 / 93.0 | 88.0 / 94.0 | 75.0 |
| hyperbaton | 96.0 / 95.0 | **99.0 / 99.0** | 94.0 / 94.0 | 95.0 / 95.0 | 98.0 / **99.0** | 98.0 / 97.0 | 93.0 |
| logical_deduction_five_objects | 58.0 / 60.0 | **66.0 / 65.0** | 52.0 / 53.0 | 59.0 / 59.0 | 57.0 / 61.0 | 53.0 / 51.0 | 56.0 |
| logical_deduction_seven_objects | 54.0 / 55.0 | **57.0 / 57.0** | 39.0 / 41.0 | 53.0 / 54.0 | 52.0 / 53.0 | 51.0 / 52.0 | 44.0 |
| logical_deduction_three_objects | 86.0 / 86.0 | **98.0 / 98.0** | 77.0 / 77.0 | 93.0 / 93.0 | 89.0 / 87.0 | 97.0 / **98.0** | 87.0 |
| movie_recommendation | 96.0 / 96.0 | 94.0 / 94.0 | 85.0 / 87.0 | 95.0 / 95.0 | **97.0 / 97.0** | **97.0** / 96.0 | 94.0 |
| multistep_arithmetic_two | 2.0 / 2.0 | 2.0 / 2.0 | 6.0 / 5.0 | 2.0 / 1.0 | 49.0 / 60.0 | **59.0 / 74.0** | 49.0 |
| navigate | 62.0 / 65.0 | 59.0 / 60.0 | 67.0 / 71.0 | 61.0 / 62.0 | **93.0 / 93.0** | 87.0 / 87.0 | 89.0 |
| object_counting | 82.0 / 84.0 | 82.0 / 82.0 | 52.0 / 52.0 | 77.0 / 78.0 | 90.0 / 90.0 | **94.0 / 95.0** | 90.0 |
| penguins_in_a_table | 64.4 / 61.0 | 71.2 / 74.6 | 62.7 / 62.7 | 59.3 / 62.7 | 72.9 / 76.3 | **78.0 / 83.1** | 57.6 |
| reasoning_about_colored_objects | 76.0 / 76.0 | 80.0 / 81.0 | 66.0 / 67.0 | 72.0 / 72.0 | 78.0 / 79.0 | **87.0 / 86.0** | 70.0 |
| ruin_names | 86.0 / 86.0 | **94.0 / 94.0** | 85.0 / 86.0 | 90.0 / 89.0 | 83.0 / 84.0 | 92.0 / 93.0 | 90.0 |
| salient_translation_error_detection | 66.0 / 63.0 | 68.0 / 68.0 | 54.0 / 54.0 | 59.0 / 64.0 | 62.0 / 66.0 | **73.0 / 74.0** | 64.0 |
| snarks | 86.1 / 86.1 | **87.5 / 87.5** | 84.7 / 84.7 | 81.9 / 86.1 | 77.8 / 79.2 | **87.5** / 86.1 | 84.7 |
| sports_understanding | 98.0 / 96.0 | **100.0 / 100.0** | 89.0 / 90.0 | 93.0 / 95.0 | 93.0 / 95.0 | 99.0 / 99.0 | 99.0 |
| temporal_sequences | 95.0 / 95.0 | **100.0 / 100.0** | 85.0 / 88.0 | 99.0 / 99.0 | **100.0 / 100.0** | **100.0 / 100.0** | 100.0 |
| tracking_shuffled_objects_five_objects | 13.0 / 13.0 | **28.0** / 26.0 | 17.0 / 15.0 | 17.0 / 16.0 | 22.0 / **27.0** | 16.0 / 17.0 | 20.0 |
| tracking_shuffled_objects_seven_objects | 17.0 / 18.0 | 11.0 / 13.0 | 7.0 / 6.0 | 16.0 / 17.0 | 18.0 / 14.0 | **23.0 / 20.0** | 20.0 |
| tracking_shuffled_objects_three_objects | 34.0 / 35.0 | 50.0 / 49.0 | 44.0 / 43.0 | 30.0 / 32.0 | 85.0 / 84.0 | **96.0 / 95.0** | 74.0 |
| web_of_lies | 51.0 / 47.0 | 99.0 / 98.0 | 48.0 / 48.0 | 48.0 / 49.0 | **100.0 / 100.0** | **100.0 / 100.0** | 100.0 |
| word_sorting | 50.0 / 49.0 | 55.0 / **55.0** | 53.0 / 53.0 | **56.0 / 55.0** | 49.0 / 48.0 | 45.0 / 48.0 | 52.0 |
| Average | 67.8 / 68.0 | 72.8 / 73.1 | 62.0 / 62.1 | 64.6 / 65.3 | 73.4 / 75.2 | **76.7 / 77.6** | 70.4 |

Table 3: BIG-Bench Hard per-task results. TRICE (CV) uses the control-variate gradient estimator with subsampling; TRICE (no CV) uses the basic gradient estimator. RS-4 is rejection sampling with four particles. GD stands for greedy-decoding accuracy (%) and SC stands for self-consistency accuracy (%).

## D  BBH Per-Task Experimental Results

Table 3 summarizes our experimental results for each task in BBH.

## E  Example Rationales

Figure 3 illustrates some examples of rationales generated by the TRICE-tuned LLM on the BBH task Logical Deduction Three Objects.

Figure 4 illustrates some examples of rationales generated by the TRICE-tuned LLM on GSM8K. Although we did find examples where the LLM got the answer right for the wrong reasons, this was much less common on GSM8K than on BBH, since the numeric output space for GSM8K is much larger than that for the typical multiple-choice BBH task.

Figure 5 shows an example where the guide model $q_\phi(z \mid x, y)$ in reweighted wake-sleep learns to closely mimic the prior model $p_\theta(z \mid x)$ until the very end of the rationale, at which point it pastes in the correct answer.

## F  Method and Template Details

In this section, we present more details on the methods and templates that we used in the experiments.

To sample from $p_\theta(z \mid x)$, we prompt the LLM with the template "Question: <QUESTION>\nAnswer: Let's think step by step.\n". We cap the length of the generated rationales at 1.25 times the length of the longest of the exemplar rationales used to initialize the soft prompt. To initialize the memory (i.e., to sample from $q(z \mid x, y)$ in line 2 of Algorithm 1), on BBH we sample from the base model with a "guide" prompt of the form "Question: <QUESTION>\nAnswer: The answer is <ANSWER>. Let's think step by step.\n". We use the same guide prompt to generate rationalizations in STaR, but with temperature 0 (see below). On GSM8K, we instead initialize the memory with samples from $p_\theta(z \mid x)$, since we found that initializing the memory using a prompt that includes the answer led to

---

**Example Rationales for Logical Deduction Three Objects BBH Task**

**An example where TRICE gets the answer right with a good rationale:**

*Question:*    On a shelf, there are three books:  a brown book, a yellow book, and an orange book.  The yellow book is the leftmost.  The orange book is to the right of the brown book.
Options:
(A) The brown book is the rightmost
(B) The yellow book is the rightmost
(C) The orange book is the rightmost

*TRICE answer:*  Brown book is to the right of yellow book.  And orange book to the right of brown book.  So.  the answer is (C).

**An example where TRICE gets the answer right with a bad rationale:**

*Question:* A fruit stand sells three fruits:  plums, apples, and loquats. The loquats are the cheapest.  The plums are less expensive than the apples.
Options:
(A) The plums are the second-most expensive
(B) The apples are the second-most expensive
(C) The loquats are the second-most expensive

*TRICE answer:*  Apples cost more than plums, but less than loquat.  So. the answer is (A).

**An example where TRICE gets the answer wrong:**
*Question:*  A fruit stand sells three fruits:  mangoes, watermelons, and kiwis.  The watermelons are less expensive than the kiwis.  The kiwis are the second-most expensive.
Options:
(A) The mangoes are the most expensive
(B) The watermelons are the most expensive
(C) The kiwis are the most expensive

*TRICE answer:* The Mangoes are more expensive than the Watermelon.  And are more expensive than kiwis.  So.  the answer is (C).

---

Figure 3: Examples of rationales where TRICE gets the answer right, right but for the wrong reasons, and wrong.



**Example Rationales for GSM8K**

**An example where TRICE gets the answer right with a good rationale:**

*Question:*   Gabrielle sells eggs.   On Monday she sells 5 crates of eggs. On Tuesday she sells 2 times as many crates of eggs as Monday.   On Wednesday she sells 2 fewer crates than Tuesday.   On Thursday she sells half as many crates of eggs as she sells on Tuesday.   How many total crates of eggs does she sell for the 4 days?

*TRICE answer:* Monday = 5 crates
Tuesday = 2 x 5 = 10 crates
Wednesday = 10 - 2 = 8 crates
Thursday = 1/2 x 10 = 5 crates
Total = 5 + 10 + 8 + 5= 28 crates
```
<answer>
28
</answer>
```

**An example where TRICE gets the answer right with a bad rationale:**

*Question:*    Samantha's last name has three fewer letters than Bobbie's last name.   If Bobbie took two letters off her last name, she would have a last name twice the length of Jamie's.   Jamie's full name is Jamie Grey.   How many letters are in Samantha's last name?

*TRICE answer:* Jamie's full name consists of 5 letters.
If Bobbie took 2 letters off her last name, she would have a last name twice the length of Jamie's, which is 10 letters.
So, Bobbie's last name has 10 letters.
If Bobbie's last name has 10 letters, and Samantha's last name has three fewer letters than Bobbie's, Samantha's last name has 10-3 = 7 letters.
```
<answer>
7
</answer>
```

**An example where TRICE gets the answer wrong:**
*Question:*    John drives for 3 hours at a speed of 60 mph and then turns around because he realizes he forgot something very important at home.   He tries to get home in 4 hours but spends the first 2 hours in standstill traffic.   He spends the next half-hour driving at a speed of 30mph, before being able to drive the remaining time of the 4 hours going at 80 mph.   How far is he from home at the end of those 4 hours?

*TRICE answer:*   First, he drives 3 hours * 60 mph = 180 miles away from home.
Then, he turns around and spends 2 hours in traffic (0 mph)
Then, he spends 0.5 hours at 30 mph = 15 miles
Then, he spends 1.5 hours at 80 mph = 120 miles
So, at the end of the 4 hours, he is 180 - 15 + 120 = 285 miles away from home
```
<answer>
285
</answer>
```



Figure 4: Examples of rationales where TRICE gets the answer right, right but for the wrong reasons, and wrong.

Figure 5: Example where the prompt-tuned RWS guide model pastes in the correct answer at the end, contradicting the rationale up to that point. The rationales generated by the guide and model are almost identical up to the final answer block.

slower convergence and worse results; it may be that including the answer in the prompt sometimes encourages the model to produce untrustworthy explanations (Turpin et al., 2023).

To evaluate the correctness $c(z, y)$ of a rationale $z$ given the answer $y$, in BBH we search the end of the rationale for final answers in the form "the answer is <ANSWER>.". In GSM8K, we initialize the soft prompt to encourage the model to wrap its answers in "<answer>" and "</answer>" tags, and then extract the answer from those tags. To encourage the bootstrapped few-shot examples in GSM8K to follow this template, we the following example to the CoT prompt: "Question: What is 1 plus 1?\nAnswer: Let's think step by step.\n1 plus 1 is 2.\n<answer>\n2\n</answer>\n\n".

Figure 6 and Figure 7 show the bootstrapped few-shot CoT examples that we used to initialize the soft prompt in the experiments.

**TRICE.** For all BBH tasks, we run TRICE for 500 steps with batch size $M = 32$ and do not use subsampling (i.e., compute $L = 64$ gradients per batch). We use the Adam optimizer (Kingma & Ba, 2015) with an initial learning rate 1.0 and a cosine decay schedule (Loshchilov & Hutter, 2017) that reduces the learning rate by 10x over the first 450 steps. For GSM8K, we run TRICE for 5000 steps with a constant learning rate of 1.0, batch size $M = 128$, and compute $L = 128$ gradients per batch.

**STaR.** We use an adaptation of the STaR strategy proposed by Zelikman et al. (2022), where we do prompt-tuning rather than fine-tuning on all weights. The method alternates between updating its memory and retuning the model from scratch on the updated memory in an inner loop. We apply this procedure for 10 outer-loop steps. Following Zelikman et al. (2022), we start with 40 inner-loop optimization steps, increasing the number of inner-loop steps by 20% each outer-loop iteration up to a maximum of 200 steps. If the training loss goes below 0.01 we break out of the inner loop. For STaR's inner-loop optimization, we use the same prompt-tuning initialization, Adam hyperparameters as above, but with cosine decay from 1.0 to 0.1 over the course of each inner loop. To update the

STaR memory, we first try generating a rationale from the prompt-tuned model by greedy decoding, then if that rationale is incorrect fall back on a rationalization generated by greedy decoding from the same guide model we use in TRICE to initialize the MCMC memory, and finally if neither procedure generates a valid rationale we omit the example from the memory.

**Rejection Sampling.** We reduce mini-batch size by $4$ and draw $4$ rationales for each example in the mini-batch. We use the same mini-batch size, train steps, and optimizer as in TRICE for all BBH and GSM8K experiments. In BBH, we use the initial learning rate $1.0$ as in TRICE. In GSM8K, we use the learning rate $0.1$ because it achieved better results than learning rate $0.3$, and the training procedure became unstable with learning rate $1.0$.

**CoT Prompt Tuning.** To do supervised CoT tuning, we prompt-tune the model to maximize the log-likelihoods of the training rationales given questions. The BBH datasets include very few exemplar rationales, so we cannot apply this strategy to BBH. On GSM8K, we use the same hyperparameters as in TRICE except that we early-stop the algorithm after only 1000 train steps, since the model overfits badly when we run longer.

**Direct Prompt Tuning.** In this method, the model tries to guess the answer directly without generating a rationale; prompt-tuning to maximize the log-likelihood of the answers in this setup is straightforward, since there is no latent rationale to integrate out. We initialize the soft prompt using 3 examples from the training set and truncate its length to 64. The optimization procedure is carried out over 150 steps with batch size 16 and the same Adam hyperparameters as above, except that the cosine decay period is set to 150 instead of 450. We found these adjustments to the hyperparameters from different training schemes were necessary to reduce overfitting.

## Bootstrapped 3-shot CoT Prompt for Formal Fallacies BBH Task

```
Question:  "It is not always easy to grasp who is consuming which
products.  The following argument pertains to this question:  First
of all, being a frequent consumer of KMS shampoo is necessary for
being a regular consumer of Yardley London soap.  Next, whoever is
a regular user of Joico shampoo is not a frequent consumer of KMS
shampoo.  All this entails that no regular consumer of Yardley London
soap is a regular user of Joico shampoo."
Is the argument, given the explicitly stated premises, deductively
valid or invalid?
Options:
- valid
- invalid
Answer:  Let's think step by step.
Regular use of a product always entails being a frequent consumer of
that product.  Although a Joico shampoo user might still buy bottles
of Yardley London soap sometimes, he/she can never be a regular
product of both soaps.  So, the answer is valid.

Question:  "Here comes a perfectly valid argument:  To start with,
whoever is an expert of BSC Young Boys is not an ex-fan of Real
Betis Balompié.  Now, whoever is a friend of FC Dynamo Kyiv is not
an expert of BSC Young Boys.  Hence, no friend of FC Dynamo Kyiv is
an ex-fan of Real Betis Balompié."
Is the argument, given the explicitly stated premises, deductively
valid or invalid?
Options:
- valid
- invalid
Answer:  Let's think step by step.
You can friend FC Dynamo Kyiv but not be an expert of BSC Young Boys.
So, the answer is invalid.

Question:  "It is not always easy to see who is related to whom -
and in which ways.  The following argument pertains to this question:
Some classmate of Terri is a workmate of Dolores.  Whoever is not a
workmate of Dolores is an ancestor of Cheryl.  So, necessarily, some
ancestor of Cheryl is not a classmate of Terri."
Is the argument, given the explicitly stated premises, deductively
valid or invalid?
Options:
- valid
- invalid
Answer:  Let's think step by step.
Reasoning:  Some classmate of Terri is not a workmate of Dolores.
Whoever is not a workmate of Dolores is an ancestor of Cheryl.  So,
necessarily, some ancestor of Cheryl is not a classmate of Terri.
Remember that Terri's classmate and Dolores's workmate can be in
any case:  One of the three can be any person.  So, the answer is
invalid.
```

Figure 6: The bootstrapped 3-shot Q-R-A soft prompt which is used in the formal_fallacies BBH experiments.

```
Question:  There are 3 meatballs on each spaghetti plate.  If Theresa's 3
sons each eat two-thirds of the meatballs on their respective plates, how
many meatballs are still left on their plates altogether?
Answer:  Let's think step by step.
There are 3x3 meat balls on each of the 3 plates.
3 sons each eat 2/3 of the meatballs on their respected plates.
<answer>
3
</answer>

Question:  Joyce, Michael, Nikki, and Ryn have a favorite movie.  Joyce's
favorite movie is 2 hours longer than Michael's movie.  Nikki's movie is
three times as long as Michael's movie, and Ryn's favorite movie is 4/5 times
as long as Nikki's favorite movie.  If Nikki's favorite movie is 30 hours
long, calculate the total number of hours of their favorite movies together.
Answer:  Let's think step by step.
Nikki's movie = 30 hours
Ryn's movie = Nikki's * 4/5 = 4/5 * 30 = 24
Michael's movie = Nikki's / 3 = 10
Joyce's movie = Michael's + 2 = 10 + 2 = 12
Total Hours = Michael's + Nikki's + Ryn's + Joyce's = 10 + 30 + 24 + 12
= 76
<answer>
76
</answer>

Question:  Susan had a bouquet of 3 dozen roses.  She gave half to her
daughter, and then placed the rest in a vase.  The next day, one-third of
the flowers in the vase were wilted.  After removing the wilted flowers, how
many flowers remained in the vase?
Answer:  Let's think step by step.
Answer STEP 1:  3 dozen is 36
Answer STEP 2:  1/2 of 36 is 18
Answer STEP 3:  18-6 = 12
<answer>
12
</answer>

Question:  Jessie won 3 times as many athletic awards as his buddy Scott,
who won 4 awards.  The best athlete at the rival high school won twice as
many awards as Jessie.  How many awards did the rival win?
Answer:  Let's think step by step.
1.  Jessie won 3 times as many awards as Scott, so Jessie won 3 x 4 = 12
awards while Scott won 4.<sentence>
2.  The best athlete at the rival school won twice as many as Jessie.  So he
won 2 x 12 = 24 awards, 12 more than Jessie, while Scott won 4.  <sentence>
3.  The best athlete at the rival school won 24 awards.  <answer>
<answer>
24
</answer>

Question:  James buys 5 packs of sodas that are 12 sodas each.  He had 10
sodas already.  He finishes all the sodas in 1 week.  How many sodas does he
drink a day?
Answer:  Let's think step by step.
There are 5 packs with 12 sodas each and he has 10 left over.
5 x 12 = 60 + 10 = 70 (number of total sodas)
70 / (7 day week) = 10 drinks per day
<answer>
10
</answer>
```

Figure 7: The bootstrapped 5-shot Q-R-A soft prompt which is used in GSM8K experiments.

