# OpenReview forum: "Training Chain-of-Thought via Latent-Variable Inference"
_NeurIPS.cc/2023/Conference — NeurIPS 2023 poster_

### Official Review · Reviewer_zJeM · 2023-07-06

**Soundness:** 3 good
**Presentation:** 3 good
**Contribution:** 3 good
**Rating:** 6
**Confidence:** 4

**Summary:**

This paper introduces a principled and practical approach to boost the training of the "chain-of-thought" through the inference of latent variables. Instead of the typical variational inference, the authors opt for a MCMC-EM method to circumvent the issue of posterior collapse, a common occurrence in auto-regressive models with latent variables, which could be exacerbated when dealing with large language models. By treating rationales as latent variables, this methodology enables training without the need for detailed rationales, which can be expensive to produce.  Moreover, the authors propose a control-variate technique that reduces training variance while maintaining unbiasedness. The theoretical underpinnings of their approach are validated on the BIG-Bench Hard dataset.

**Strengths:**

1. Although the soft-prompt approach is widely used, this proposal introduces an innovative perspective on 'prompt-finetuning' utilizing a chain-of-thought.

2. The assumptions and derivations presented are straightforward and intuitive with only minor confusion.

3. The authors thoroughly explore the links between their work and previous studies, while also critically examining the limitations of their own findings.

4. The authors validate their theories by applying them to the significant benchmark of BBH.

**Weaknesses:**

See questions.

**Questions:**

1. For accept rate in Metroplis-Hastings, why the transition kernel $p(\tilde{z},y|x)$ is not conditional on previous state $z$?

2. For sampling from intractable posterior $p(z|x,y)$, what if you directly sampling from this distribution using finite-step MCMC as $p(z|x,y)\propto p(z|x)p(y|z,x)$? You do not need to add more experiments. Could you explain the difference?

3. For control variate, in appendix you mentioned it may not be that useful in the stochastic setting. For figure 1, I want to know that suppose you train long enough, will the training estimate becomes the same for two settings (with or without control variate) based on your claim in Eq. 3? Meanwhile, what's the notation meaning for expectation over MCMC?

4. How many tokens you use for latent variable $z$?

5. While I commend the authors for validating their approach on an important benchmark, an aspect I deeply appreciate, I believe that testing on a more diverse dataset would bolster the paper's credibility and appeal. This is particularly true when making a comparison with [1].

6. Do you have failure case analysis and do you need human evaluation?

I would be more than willing to raise my review scores if the authors could provide further clarification and deeper insights about the paper.

[1] STaR: Self-Taught Reasoner Bootstrapping Reasoning With Reasoning

**Limitations:**

The authors has discussed this paper's limitations.

---

> ### Author Rebuttal · Authors · 2023-08-09
>
> Thank you for your thoughtful questions! We will try to answer them below; please let us know if anything remains unclear or if there are particular points that you would like to see emphasized in the final paper.
>
> # 1. Acceptance probability depends on previous state
>
> Our use of a binary (deterministic) likelihood and identifying the proposal $q(z\mid x, y)$ with the prior $p(z\mid x)$ dramatically simplifies the acceptance probability, so that we only need to consider whether or not the proposed state leads to a correct answer when deciding whether or not to accept it. If we used a “soft” likelihood function as discussed in Appendix A, we would indeed need to look more closely at the previous state to compute the acceptance probability.
>
> # 2. Why not use finite-step MCMC?
>
> The main reason we only take a single MCMC step per iteration is to minimize the per-update cost of the algorithm; the cost of drawing a sample is comparable to the cost of a gradient update for the prompt, so it seems natural to balance these costs.
>
> # 3. Is the control variate necessary late in training?
>
> If anything, the control variate is probably most helpful late in training, since the correlation between the control variate and the base gradient estimator increases with the accuracy of the model. There may be subtle effects on what kinds of local optima the two gradient estimators tend to yield, but these might be reduced by using a smaller learning rate and more training steps.
>
> # 3. What’s meant by $\mathbb{E}_\textrm{MCMC}$?
>
> The expectation is with respect to the randomness in the MCMC proposal, which for TRICE is just the proposed sample from $p_\theta(z\mid x)$. We will change the notation to make this clearer.
>
> # 4. How many tokens per rationale?
>
> We cap the number of tokens per rationale at 1.25 times the length of the longest of the three exemplar rationales used to initialize the soft prompt. The goal is to ensure that we have plenty of space to generate good rationales without unnecessarily increasing the cost of sampling and gradient evaluation.
>
> Thank you for pointing out that this information is missing from the paper! We’ll include it in the next version.
>
> # 5. Other datasets.
>
> See the comment at the top of the page for results on GSM8K.
>
> # 6. Failure cases.
>
> We will add examples of rationales that lead to both correct and incorrect answers in an appendix.
>
> # 6. Human evaluation.
>
> Formal human evaluation is beyond the scope of this paper, but will be important in the future in order to determine to what extent models trained using TRICE get answers right for the right reasons. We will add some discussion of this point to the Limitations section.

---

> > ### Comment · Reviewer_zJeM · 2023-08-18
> >
> > Thank you for your thoughtful response. I appreciate your feedback on my previous concerns and agree with Reviewer AjuJ about the need to enhance the notations in this paper.
> >
> > While I understand the paper utilizes a proprietary model, it would be greatly beneficial if an open-source counterpart could be provided.

---

> > > ### Author Response · Authors · 2023-08-18
> > > **reference implementation**
> > >
> > > (Largely copy-pasted from response to Reviewer d8Yh)
> > >
> > > We will open-source an IPython notebook with a reference implementation of TRICE that is agnostic to the backend LLM. To get it to work with a new pretrained LLM, the user will have to provide four callables with the following signatures:
> > > ```
> > > sample(params, context, num_steps, seed)
> > > log_prob(params, context, continuation)
> > > grad(params, context, continuation)
> > > init(seed)
> > > ```
> > > `init` should return an initial value for a PyTree `params` ($\theta$ in the notation of the paper). `sample` should draw a sample `continuation` ($z$ in the notation of the paper) from the fine/prompt-tuned LLM of maximum `num_steps` tokens given `context` ($x$ in the notation in the paper). `log_prob` should compute the log-probability of generating `continuation` given `context` and `params` ($\log p_\theta(z\mid x)$), and `grad` should be the gradient of `log_prob` w.r.t. `params` ($\nabla_\theta\log p_\theta(z\mid x)$). `context` and `continuation` should be Python strings, which may incur a bit of a performance hit with unnecessary tokenization and detokenization but simplifies the API.
> > >
> > > If the user provides these four functions, the notebook will run a short training loop of TRICE on a BBH task. Hopefully this will make it easier for others to reproduce our results using their own LLMs (whether open-source or proprietary).

---

### Official Review · Reviewer_AjuJ · 2023-07-06

**Soundness:** 3 good
**Presentation:** 3 good
**Contribution:** 3 good
**Rating:** 6
**Confidence:** 4

**Summary:**

This paper explores considering Chain-of-Thoughts (CoT) as a latent variable and introduces TRICE: an MCMC-EM algorithm for optimizing CoT-latent-latent variable models using a Metropolis-Hastings algorithm (rejection sampling) coupled with a control variate.

The authors tested the model by fine-tuning a proprietary LLM on 27 BIG-bench hard tasks and report improvements when compared to direct prompt tuning and STaR. The authors tested the effectiveness of the control variate but did not test the method against more conventional optimisation methods.

**Strengths:**

Overall, This paper discusses a well-defined issue and explores innovative solutions to the optimization problem, it was a refreshing and educative read, thank you!

Main strengths:
1. Optimizing LLMs as latent variable models has great potential and the authors present the problem well
2. The authors introduce an innovative gradient estimator (MCMC-EM + control variate) which is computationally efficient (single sample per data point) and seems to work very well
3.  The authors demonstrate progress on a subset of the MMLU dataset
4.  Theoretical and experimental limitations are thoroughly discussed

**Weaknesses:**

Overall, the paper lacks structure and more extensive comparison with the existing literature on Monte Carlo methods. I believe not addressing these points would greatly reduce the impact of this paper. Nevertheless, I would happily accept this paper if these limitations are resolved!

1. The paper overall lacks structure. Although I appreciated the lengthy discussions, I think the main storyline could be made more clear by separating the main ideas from the discussions.

2. The optimization method is only presented using full text and Algorithm 1. This paper would convey its idea more clearly using a set of two additional equations: A) expression of the *true* gradients and B) Monte Carlo estimate (which translates into Algorithm 1).

3. Classic gradient estimation methods are not covered sufficiently and some of the statements in section 2.2 *Why not variational inference?* are misleading.
	1. Posterior collapse (discussed in Lucas et al., 2019) is an issue inherent to the optimization of the proposal distribution $q(z | x, y)$ and $q$ is in this paper never optimized.
	2. The author discusses the limitations of the ELBO, but the ELBO only arises when using a single-sample estimate of the log-likelihood. When working with discrete latent variables, it is more common to use the Importance Weighted Bound (IWB). I.e., use a K-samples importance-weighted estimate of the likelihood (e.g., $\log p(y | x) \approx \log \frac{1}{K} \sum_{j=1}^K \frac{p(y, z^{(j)} | x)}{q(z^{(j)}|x,y)})$ -- as done in ["Revisiting Reweighted Wake-Sleep for Models with Stochastic Control Flow", Let et al., 2019](https://arxiv.org/abs/1805.10469)) which variance scales as $\sqrt{1/K}$. See ["Monte Carlo Gradient Estimation in Machine Learning", Mohamed et al., 2020] for more background on optimization methods.

6. The experimental comparison with RWS is not included in the paper.

**Questions:**

1. How exactly did you compute the gradients in the RWS experiment?

**Limitations:**

1. I stumbled upon this PhD thesis: [Deep Latent Variable Models for Natural Language Processing, Lievin 2022](http://vlievin.github.io/deep-lvms-for-nlp.pdf), which discusses similar ideas. You might not have been the first ones to consider that "CoT methods are latent-variable probabilistic models"

---

> ### Author Rebuttal · Authors · 2023-08-09
>
> Thank you so much for the thoughtful questions and suggestions. We hope that our response below will address your concerns.
>
> # 1. Structural improvements
>
> This is a great suggestion. To reduce the risk that readers will lose the main thread, in the final version we will more clearly set apart as “remarks” discussions that are meant to provide context and intuitions.
>
> # 2. Providing true gradient and Monte Carlo estimate
>
> The true gradient of the per-example log-likelihood is given in Equation 2, and the Monte Carlo estimate is given in Equation 3 and expanded in Equation 4. Perhaps we’re misunderstanding your suggestion?
>
> # 3, 4: VI, RWS, and the IWB
>
> Thank you for pointing out some ways in which we can make this section clearer. We have rewritten section 2.2 to focus more on RWS and the IWB (which we also tried before settling on MCMC-EM; see below) than on VI. Hopefully the version below is an improvement—we’d be very grateful for any further feedback!
>
> ## 2.2 Why not variational inference, reweighted wake-sleep, or the importance-weighted bound?
>
> We considered three alternatives to the MCMC-EM approach that we pursue in this paper: variational EM (e.g., Bishop, 2006), reweighted wake-sleep (RWS; Bornschein & Bengio, 2015; Le et al., 2019), and the importance-weighted bound (IWB) used in importance-weighted autoencoders (IWAE; Burda et al., 2015).
>
> Variational EM is a common strategy for training latent-variable models, but variational inference with discrete latent variables is challenging (e.g., Tucker et al., 2017). [pointer to VI discussion?] RWS is an attractive alternative that avoids high-variance score-function gradients; it proceeds by sampling $M$ samples $z_{1:M}$ from a guide model $q_\phi(z\mid x, y)$, assigning the samples weights $w_m\propto \frac{p_\theta(y, z\mid x)}{q_\phi(z\mid x)}$, and updating both the model parameters $\theta$ and the guide parameters $\phi$ to maximize the reweighted log-probabilities $\sum_m w_m \log p_\theta(z_m\mid x)$ and $\sum_m w_m \log q_\phi (z_m\mid x)$. Unfortunately, we found that RWS training sometimes produced models that produced degenerate zero-length rationales $z$. Digging deeper, we found that the weights $w_m$ tend to be larger for shorter sequences $z_m$, so the model and guide learn to produce shorter and shorter sequences until they consistently produce empty rationales.
>
> Why do longer sequences tend to get lower weights? We can write the unnormalized weights as $\tilde w_m = (c(y, z_m) + \epsilon)\frac{p_\theta(z_m\mid x)}{q_\phi(z_m\mid x)}
> = c(y, z_m)\prod_{t=1}^{T_m} \frac{p_\theta(z_{m,t}\mid x, z_{m,1:t-1})}{q_\phi(z_{m,t}\mid x, z_{m,1:t-1})}$,
> where $T_m$ is the length of $z_m$, and $\epsilon$ is added to address the case where none of the $M$ samples are correct. If there is a mismatch between $q(z_{m,t}\mid x, z_{m,1:t-1}))$ and $p(z_{m,t}\mid x, z_{m,1:t-1}))$, then $\frac{p_\theta(z_{m,t}\mid x, z_{m,1:t-1})}{q_\phi(z_{m,t}\mid x, z_{m,1:t-1})}$ will typically be less than one, with a few rare exceptions with extremely high weights ensuring that $\mathbb{E}_q[p(z\mid x)/q(z\mid x)]=1.$
>
> If these exceptions are rare enough that they do not typically appear in a sample of $M$ sequences $z_{1:M}$, then the normalized weights $w_{1:M} = \frac{\tilde w_{1:M}}{\sum_m \tilde w_m}$ will tend to assign higher mass to shorter sequences unless those shorter sequences are much less likely to be correct.
>
> Finally, we considered optimizing the $M$-sample IWB
>
> $\mathcal{L}(x, y, \theta) \triangleq \mathbb{E}_{p}[\log \frac{1}{M}\sum_m\frac{p(y, z\mid x) + \epsilon}{p(z\mid x)}]$
>
> (where we use $p_\theta(z\mid x)$ as an importance-sampling proposal, $M$ is the number of draws from $p_\theta(z\mid x)$ and $\epsilon$ is added to avoid taking the log of zero). This works well as long as the number of samples per example $M$ is large enough, but sampling several rationales per example carries a high per-step cost; the independence-chain MCMC strategy we use in TRICE lets us do $M$ times as many parameter updates per sample from $p_\theta(z\mid x)$.
>
> > How exactly did you compute the gradients in the RWS experiment?
>
> We draw $M$ samples $z_{1:M}$ from a guide model $q_\phi(z\mid x, y)$, compute the unnormalized weights
>
> $\tilde{w}_m = \frac{p(z\mid x)\epsilon^{(1 - c(y,z))}}{q(z\mid x,y)},$
>
> in the log-space, assign the weights
>
> $w_m = \mathrm{stopgradient}(\tilde{w}_m / \sum_m\tilde{w}_m),$
>
> and optimize the loss $\sum_m w_m \left(-\log p_\theta(z_m\mid x) - \log q_\phi(z_m\mid x,y)\right)$. This objective gives us an estimation
> $\sum_m -w_m \nabla_\theta \log p_\theta(z_m\mid x)$ for the gradient w.r.t. $\theta$ and  $\sum_m -w_m \nabla_\phi \log q_\phi(z_m\mid x,y)$ for the gradient w.r.t. $\phi$.
>
> The $\epsilon^{(1 - c(y,z))}$ factor represents the likelihood $p(y\mid x,z)$. The likelihood of arriving at the correct answer given a correct rationale is 1 and the likelihood of arriving at the correct answer given a wrong rationale is $\epsilon$. This has the same effect as adding $\epsilon$ to the binary likelihood that we described above: to avoid taking the log of zero when all rationales are wrong. In the experiments, we set $\epsilon = 0.001$.
>
> # Lievin thesis
>
> Thanks for the reference to Lievin thesis! We will cite it in the revision. Although that work describes a latent-variable chain-of-thought model, we note that the inference methods used for the COT model are relatively simple (in contrast to other models from the thesis), similar in spirit to self-consistent COT. Our work exploits the connection to probabilistic modeling to apply more sophisticated inference algorithms; we are not aware of previous work that takes this strategy. In contrast, previous work focuses on simple Monte Carlo (self-consistent COT, Lievin 2022), or rejection sampling (STaR).

---

> > ### Comment · Reviewer_AjuJ · 2023-08-14
> > **Improving clarity and presenting main equations more clearly**
> >
> > ### 1. Structural improvement
> > Enclosing remarks would indeed help.
> >
> > ### 2. Providing true gradient and Monte Carlo estimate
> >
> > Yes, the true gradient is provided in Eq 2. However, the expression of the gradient approximation is spread across sections "**2.1 derivations**" and Eq 3&4. Equations 3&4 (which are the core equations defining TRICE) are enclosed in a section titled "**6 Adding a control variate**". A better structure, better choice of section titles and an improved framing of the equations would help.
> >
> > 1. Gradient
> > $$\eta_\theta = E_{p_\theta(z | x, y)} \left[  \nabla_\theta \log p_\theta(z | x) \right]$$
> > 2. Gradient approximation via MCMC
> > $$\eta_\theta \approx E_{\mathrm{MCMC}} \left[  \nabla_\theta \log p_\theta (z | x) \right] $$
> > 3. Gradient approximation via MCMC with control variate
> > $$\eta_\theta \approx \ldots $$
> >
> > **NB**: Similarly to Reviewer zJeM, I find the notation $E_{\mathrm{MCMC}}\left[ \cdot \right]$ confusing.

---

> > > ### Author Response · Authors · 2023-08-15
> > > **Thanks for the suggestions!**
> > >
> > > Thanks for expanding on your suggestions about improving the presentation of the gradient estimators! We very much appreciate your taking the time to help us improve the paper's presentation.
> > >
> > > We agree that a structure along the lines you suggest will be clearer: we will explicitly delineate the path from
> > > 1. true-but-intractable gradient to
> > > 2. MCMC estimate to
> > > 3. variance-reduced MCMC estimate.
> > >
> > > And we agree that the $\mathbb{E}_\mathrm{MCMC}$ notation is a bit ambiguous. We will rework it to something more explicit (and closer to what's in Algorithm 1), something along the lines of
> > >
> > > $z' = c(\\tilde z, y)\\tilde z + (1-c(\\tilde z, y))z; \\ \\hat g = \\nabla_\\theta\\log p_\\theta(z'\\mid x); \\ \\mathbb{E}_{z,\\tilde z}[\\hat g\\mid\\theta] \\approx \\mathbb{E}\_{p\_\\theta(z\\mid x, y)}[\\nabla\_\\theta \\log p\_\\theta(z\\mid x)],$
> > >
> > > where $\mathbb{E}\_{z,\tilde z}[\cdot\mid\theta]$ denotes an expectation with respect to both the proposal $\tilde z$ and the previous state $z$.
> > >
> > > This in turn lets us write the derivation of the control-variate estimator as
> > >
> > > $\mathbb{E}\_{z,\tilde z}[\hat g\mid \theta] = \mathbb{E}\_{z}[\mathbb{E}\_{\tilde z}[\nabla\_\theta\log p\_\theta(z'\mid x)] \mid\theta]$
> > >
> > > $= \mathbb{E}\_{z}[\mathbb{E}\_{\tilde z}[\nabla\_\theta\log p\_\theta(z'\mid x) - \beta\nabla\_\theta\log p\_\theta(\tilde z\mid x)] \mid\theta]$
> > >
> > > which is hopefully also a bit clearer.

---

> > ### Comment · Reviewer_AjuJ · 2023-08-14
> > **VI, RWS, and the IWB - great insights**
> >
> > Thank you for the clarification. It makes sense you obtain zero-length rationales, and I understand why you relate it to posterior collapse.
> >
> > > How exactly did you compute the gradients in the RWS experiment?
> >
> > Thank you for the details. What are your thoughts on optimizing $\theta$ using the "M-sample IWB" only? Meaning, using the expression of the RWS gradients you provided and using, for instance, $p(z|x,y) \approx p_\theta(z|x) c(x,y)$ as sampling distribution? How does it relate to TRICE?

---

> > > ### Author Response · Authors · 2023-08-15
> > > **M-sample IWB optimization**
> > >
> > > We did try an M-sample IWB strategy (equivalent to RWS with a fixed $q(z\\mid x, y) = p\_\\theta(z\\mid x)$), and found that it worked well as long as the number of samples $M$ was large enough. The main advantage of TRICE over this IWB approach is that it lets us draw $M$ times fewer samples per update, which allows for a significant improvement in wall-clock time even for relatively small values of $M$. (A secondary advantage is that TRICE frees us from having to choose an appropriate value of $M$ that balances bias and computational cost.)
> > >
> > > If we do multiple MCMC updates in TRICE before computing a gradient update, it begins to look more like the variant of IWB optimization where we do $M$ forward passes and randomly choose a single particle to compute the backwards pass to save computation (as suggested at the end of section 3 in the 2015 IWAE paper). The $M$ proposals can be generated and evaluated in parallel, and the state we compute gradients on is the "last" one to be accepted ("last" in scare quotes since the order is arbitrary). The only difference is the case where none of the $M$ particles yield correct answers, in which case TRICE still has the option of falling back on a state generated in a previous iteration.

---

> ### Comment · Reviewer_AjuJ · 2023-08-16
> **Increasing my rating 4 -> 6**
>
> Thank you for providing more details about your experiments with the IWB. Based on our discussion, I understand that you have experimented with many objectives, including multi-sample objectives (IWB).
>
> Trusting that you will include more details in the paper (comparison with a vanilla IWB objective) and that you will improve the structure of the paper (equations), I am increasing my rating as follows:
>
> - **presentation**: 2 -> 3
> - **soundness**: 2 -> 3
> - **rating**: 4 -> 6
>
> I would be willing to increase my rating further if the experiment section was to be expanded to include
> - a comparison with the IWB and/or other objectives you have experimented with.
> - insights on the variance of TRICE compared to other baselines

---

> > ### Author Response · Authors · 2023-08-18
> > **IWB results**
> >
> > Thank you for being so engaged in the discussion process!
> >
> > We ran (and will include in the final paper) an additional set of BBH experiments (with the higher-quality base model discussed above) using the IWB with 4, 8, and 16 particles and the bootstrapped initial parameters. The results (with TRICE for comparison) are:
> >
> > |Eval Method|4\-Sample IWB|8\-Sample IWB|16\-Sample IWB|TRICE|
> > |-|-|-|-|-|
> > |Greedy Decoding|64.6|67.0|66.2|72.8|
> > |Single CoT Sample|64.2|65.6|64.6|72.5|
> > |Self Consistency (40 CoT Samples)|65.3|67.4|66.6|73.1|
> >
> > We will also include a figure demonstrating the qualitative behavior of RWS.

---

### Official Review · Reviewer_SKio · 2023-07-07

**Soundness:** 4 excellent
**Presentation:** 4 excellent
**Contribution:** 3 good
**Rating:** 7
**Confidence:** 4

**Summary:**

This work introduces a new method for training models for chain-of-thought prompting, aimed at maximizing the marginal probability of generating a correct answer using Markov-chain Monte Carlo, expectation maximization, and a novel control variate technique, along with prompt tuning. Much of the work is spent deriving and analyzing the method (e.g. compared to related methods like self-taught reasoner STaR), with a final section dedicated to testing the method and demonstrating very significant performance gains over vanilla prompt tuning (no CoT) and STaR.

**Strengths:**

- Principled, novel method along with extensive derivation. Particularly, this well motivates the need for taking a marginalization framing rather than the greedy approach often used before
- It is generally very useful to see more theoretical justification provided in this area. Past works (e.g. STaR) provide many novel ideas, but it is useful to take a more comprehensive (and marginalization-based) approach to the problem
- extensive theoretical/technical rationale for method over other options (STaR, variational inference)
- very strong experimental results, demonstrating significant gains over STaR, another recent method for training for CoT prompting

**Weaknesses:**

- Fairly limited analysis of experimental results (mostly aggregate results on Big Bench tasks in the main paper, without too much specific analysis)
- It would also be useful/interesting to see what the generated rationales look like: how do they differ from e.g. STaR, and how often are they correct in a human-interpretable way rather than just leading to a correct answer? Adding some information about this in the main paper would be useful
- Fairly limited discussion of limitations

**Questions:**

None

**Limitations:**

Limitations section is quite limited. It would be useful to see e.g. a discussion of the effects of rationales on user trust (will having rationales make users trust models more even if the rationales are wrong?)

---

> ### Author Rebuttal · Authors · 2023-08-09
>
> Thank you for the positive feedback!
>
> # Additional experimental results
>
> While there is limited space in the main text, we will add an appendix with a full table of per-task BBH results.
>
> Also, we added an experiment on GSM8K; see the top comment for details.
>
> ### Qualitative analysis of rationales
>
> Thank you for this suggestion—we will add some example rationales (both correct and incorrect) from TRICE and STaR to the final version.
>
> # Discussion of user trust
>
> Thank you for your suggestion that we add some discussion of the impact of rationales on user trust to the limitations section. On the one hand, providing rationales may make it easier for motivated users to judge the trustworthiness of LLMs for themselves; if an LLM's reasons are bad, we should consider its conclusions suspect. On the other hand, many users may not bother to read and critique an LLM's rationales, and may take a rationale's mere existence as evidence of correctness. While the focus of this work is on improving LLM capabilities, we should at least acknowledge that much work remains to be done on alignment questions such as this.

---

> > ### Comment · Reviewer_SKio · 2023-08-15
> >
> > Thank you for the response! I look forward to this further information in later versions of the paper

---

### Official Review · Reviewer_d8Yh · 2023-07-07

**Soundness:** 3 good
**Presentation:** 3 good
**Contribution:** 3 good
**Rating:** 5
**Confidence:** 5

**Summary:**

This paper proposes treating rationales as latent-variables and considers the marginal distribution over answers, averaging over possible rationales. To do so, a MCMC procedure is proposed, in which rationales are proposed via an independence sampler. The approach is compared to STaR (Zelikman et al., 2022) as a baseline on 27 BigBench-HARD tasks. Rather than fine-tune all the model weights, prompt tuning is used a parameter-efficient alternative. The proposed approach is found to outperform STaR on the subset of tasks considered.

**Strengths:**

* The derivations the authors present are detailed and comprehensive. The design choices are explained well and justified in context.

* Compared to alternate methods, the approach advocated in this paper seems more principled.

* The authors discuss a potential alternate approach (variational Bayes) and justify why it is not appropriate.



**Weaknesses:**

* As the authors acknowledge, TRICE is only tested on small datasets. The behavior of the comparison model is significantly different than the expected behavior; the authors speculate this is due to overfitting of the comparison model, but do not compare TRICE’s performance on Commonsense QA or reduce the number of inner loop steps to prevent overfitting.

* While the overall approach is novel, it’s not clear that the improvements and alterations proposed present significant innovations on existing methods. Furthermore, I have some concerns about the sensitivity of the approach to hyper-parameters, such as the # of MCMC steps, initialization scheme, etc.

* Some implementation details are unclear. The methods as-is would not be reproducible, and it seems that the authors do not plan to release their code, raising concerns about replicability.

**Questions:**

* What is the specific $c(z,y)$ used in the experiments? It never seems to be defined— an example is given on line 71, but it’s unclear whether that’s the specific correctness function used in the experiments.

* Line 80 says that “For example, the guide might prompt an LLM specifically to give an rationale for the answer”— was this the case for all experiments?

* Line 296 says that "an adaptation of the STaR strategy" is tested-- what aspects specifically were adapted?

**Limitations:**

There is a nominal discussion of limitations but I think this could be significantly expanded. For example, MCMC can be sensitive to hyperparameters and mixing is hard to diagnose.

---

> ### Author Rebuttal · Authors · 2023-08-09
>
> Thank you so much for the thoughtful questions and suggestions. We hope that our response below will address your concerns.
>
> # Larger datasets
>
> We added an experiment on GSM8K; see the top comment for details.
>
> # Sensitivity to number of MCMC steps
>
> In Algorithm 1 (and in all of our experiments) we just do a single MCMC update per iteration. It might be interesting to explore treating the number of MCMC steps per iteration as a hyperparameter, but we feel that 1 is a natural default that allows for the prompt to be updated as often as possible. (This is somewhat analogous to the common practice of training VAEs using single-sample estimates of the per-example loss.)
>
> # Sensitivity to initialization
>
> Prompt tuning has been known to be sensitive to initialization from its inception; for example, random initializations tend to underperform initializations based on embeddings of real text sequences (e.g., [3]). We also found that initializing the soft prompt with human-generated rationales (rather than bootstrapped rationales) does slightly improve TRICE’s performance (Table 1), but both initialization schemes worked well.
>
> # Reproducibility
>
> Since our code uses a proprietary model, we cannot release the code we used in our experiments. We tried to include enough information in the main text for a reader to reimplement these experiments, but we may have missed a few details—we will add an appendix with pseudocode describing in more detail all of the algorithms that we ran, which may be a clearer format for providing this information than the dense text blocks in Section 4. We appreciate your identifying some specific points that were not clear in the “questions” section above (which we will address below)—if you notice any other details that are missing from the main text please do let us know!
>
> # MCMC hyperparameters and mixing diagnostics
>
> While MCMC kernels in general are often sensitive to hyperparameters and difficult to diagnose, the particular independence-chain proposal we use in TRICE has no hyperparameters (it simply makes proposals from the prior) and, in the binary-likelihood setting we consider, generates unbiased samples as soon as it accepts a proposal (since it is effectively doing rejection sampling on the posterior). We will emphasize these points in the final version.
>
> # Answers to specific questions
>
> > What is the specific about c(z, y) used in the experiments?
>
> Thank you for pointing out that this was not precise. We compare the final tokens of $z$ to the tokens of the target “the answer is {y}.”. This is to account for various formats of the answers like “So the answer is (A).” or “Thus the answer is (A).” We will clarify this in the paper.
>
> > Line 80 says that “For example, the guide might prompt an LLM specifically to give a rationale for the answer”— was this the case for all experiments?
>
> Yes, we used the same prompt template across all experiments. We described the template in the Experiments section (lines 267-272).
>
> > Line 296 says that "an adaptation of the STaR strategy" is tested-- what aspects specifically were adapted?
>
> In the STaR paper, the authors performed fine-tuning of all LLM parameters while in our work, we use a parameter-efficient tuning method, namely prompt tuning. We will clarify this in the paper.
> \
> \
> \
> [3] Li, Xiang Lisa, and Percy Liang. "Prefix-tuning: Optimizing continuous prompts for generation." _arXiv preprint arXiv:2101.00190_ (2021).

---

> > ### Comment · Reviewer_d8Yh · 2023-08-16
> >
> > Thank you for the response, which has clarified some details that were unclear in the initial submission. Although the code used in the experiments may be proprietary, this doesn't preclude producing a separate reference open-source implementation (perhaps with reduced capabilities, working with open-source LLMs, etc.).

---

> > > ### Author Response · Authors · 2023-08-18
> > > **Reference implementation**
> > >
> > > > Although the code used in the experiments may be proprietary, this doesn't preclude producing a separate reference open-source implementation...
> > >
> > > This is an excellent suggestion! We will open-source an IPython notebook with a reference implementation of TRICE that is agnostic to the backend LLM. To get it to work with a new pretrained LLM, the user will have to provide four callables with the following signatures:
> > > ```
> > > sample(params, context, num_steps, seed)
> > > log_prob(params, context, continuation)
> > > grad(params, context, continuation)
> > > init(seed)
> > > ```
> > > `init` should return an initial value for a PyTree `params` ($\theta$ in the notation of the paper). `sample` should draw a sample `continuation` ($z$ in the notation of the paper) from the fine/prompt-tuned LLM of maximum `num_steps` tokens given `context` ($x$ in the notation in the paper). `log_prob` should compute the log-probability of generating `continuation` given `context` and `params` ($\log p_\theta(z\mid x)$), and `grad` should be the gradient of `log_prob` w.r.t. `params` ($\nabla_\theta\log p_\theta(z\mid x)$). `context` and `continuation` should be Python strings, which may incur a bit of a performance hit with unnecessary tokenization and detokenization but simplifies the API.
> > >
> > > If the user provides these four functions, the notebook will run a short training loop of TRICE on a BBH task. Hopefully this will make it easier for others to reproduce our results using their own LLMs (whether open-source or proprietary).

---

### Official Review · Reviewer_Surj · 2023-07-10

**Soundness:** 3 good
**Presentation:** 2 fair
**Contribution:** 2 fair
**Rating:** 4
**Confidence:** 4

**Summary:**

The paper proposes a prompt-tuning strategy that tries to maximize the marginal log-likelihood of generating a correct answer using CoT prompting.

**Strengths:**

The paper presents a prompt-tuning strategy that aims to optimize the marginal log-likelihood of producing accurate answers through the utilization of CoT prompting. And the paper uses  simple Markov chain Monte Carlo (MCMC) expectation-maximization (EM) algorithm, memoized wake-sleep, and persistent contrastive divergence to address the challenge of sampling from the posterior. The results look like promising over other baselines.

**Weaknesses:**

The author did not provide specific information about the model details. If the model size is close to 100 billion, it would be beneficial for the paper to include a baseline comparison with text-davinci-003 (Zero-shot or few shot learning). Otherwise, I feel it unnecessary to conduct prompt tuning on a large-scale model if its performance is still inferior to that of zero/few-shot learning using the same scale model but from GPT family.
If the author's intention is to compare with open-source models, it is essential to include Alpaca-CoT (https://github.com/PhoebusSi/alpaca-CoT), regardless of the specific model or fine-tuning technique they employ.

**Questions:**

See weakness part.

**Limitations:**

See weakness part.

---

> ### Author Rebuttal · Authors · 2023-08-09
>
> Thank you for your suggestion to compare against text-davinci-003. Our model size is in the 40–75 billion parameter range, making it significantly smaller than text-davinci-003’s 175B parameters. We will add published text-davinci-003 BIG-Bench Hard (BBH) 3-shot results to Table 1 for comparison (these results are taken from [2]). The average accuracy results are summarized below:
>
> | text-davinci-003 (3-shot COT) | No tuning (3-shot COT) | TRICE (3-shot COT) | TRICE (bootstrapped COT) |
> |-|-|-|-|
> | 70.7 | 44.6 | 72.2 | 70.1 |
>
> We used the same base model to obtain results for the last 3 columns. The table above demonstrates that even though we begin with a smaller, weaker model than text-davinci-003, by doing TRICE prompt tuning, we can roughly match (if bootstrapping an initialization) or slightly improve on (if using a few-shot human-generated CoT initialization) the performance of the larger model.
>
> Also, as we discuss in the comment at the top of the page, we reran the TRICE BBH experiments using a higher-quality pretrained model, which improved our bootstrapped-initialization results from 70.1% to 72.8%.
> \
> \
> [2] “Scaling Instruction-Finetuned Language Models”, Chung et al., 2022, https://arxiv.org/pdf/2210.11416.pdf, table 19

---

### Author Rebuttal · Authors · 2023-08-09

We sincerely thank the reviewers for their many helpful and thoughtful comments.

Most of our responses are in the per-review comments, but we want to highlight here two sets of new experimental results that we will add to the paper.

# BIG-Bench Hard results with a stronger base model

After our initial submission, we reran the TRICE BBH experiments using bootstrapped initialization and a higher-quality pretrained model of similar size. Like our original base model, this model is in the 40–75B range*, significantly smaller than GPT-3's 175B parameters.

When prompt-tuning this improved model, TRICE gets an average greedy-decoding accuracy on BBH of 72.8%, significantly better than our previous 70.1%.

# GSM8K results

After our initial submission, we ran the algorithms on the GSM8K dataset [1], which includes 7,473 training examples and 1,319 testing examples, using the same higher-quality pretrained model discussed above. Below is a summary of the results with bootstrapped-initialization CoT (except for direct tuning):

| Direct tuning | No tune COT | STaR | SFT | SFT (with SC) | TRICE | TRICE (with SC) |
|-|-|-|-|-|-|-|
| 5.8 | 22.4 | 40.5 | 56.3 | 73.1 | 60.0 | 75.6 |

SFT: Supervised fine-tuning. The training dataset includes example rationales for each question; we prompt-tune the model using these pairs of questions and rationales.
SC: Self-consistency majority voting using 40 samples.

STaR performs better than direct tuning on this task, but still underperforms compared to TRICE and SFT. On the other hand, TRICE outperforms SFT even though TRICE does not train on human-generated rationales. We conjecture that this may be because SFT with cross-entropy loss gives more weight to the style of the rationales than to their content, whereas the marginal log-likelihood loss that TRICE minimizes is insensitive to precise wording.
\
\
\* We originally disclosed a larger range of 15–100B parameters, but proprietary models in this smaller 40–75B range have been publicly disclosed by many organizations such as Anthropic, Bloomberg, DeepMind, Google, Meta, and Stability, so we feel comfortable providing this narrower range.

[1] Cobbe, Karl, et al. "Training verifiers to solve math word problems." _arXiv preprint arXiv:2110.14168_ (2021).

---

### Comment · Area_Chair_ySUk · 2023-08-17

Reminder: the author-reviewer discussion is closed on Aug 21st 1pm EDT.

---

### Decision · Program_Chairs · 2023-09-21

**Decision:**

Accept (poster)

**Comment:**

The paper treats chain-of-thought-prompted question-answering LLMs as probabilistic latent-variable models, so we can derive a principled, simple, effective way of tuning them to generate better rationales without training on human-generated rationales. Most reviewers tend to accept the submission although one of them tends to reject it. Reasoning of LLMs is important for the current models. The paper proposes an interesting perspective to view the COT solution. Some LLM researchers would be interested in the proposed method. Moreover, the author response can be included in the revised version.